# CLAWS:Creativity detection for LLM-generated solutions using Attention Window of Sections

Keuntae Kim[1]*    Eunhye Jeong[2]*    Sehyeon Lee[3]    Seohee Yoon[1]    Yong Suk Choi[1]†

[1]Department of Computer Science
[2]Department of Artificial Intelligence
[3]Department of Future Mobility
Hanyang University, Seoul, Korea
{ktkpv94, jeh0826, leesehyeon, seohee09}@hanyang.ac.kr

## Abstract

Recent advances in enhancing the reasoning ability of Large Language Models (LLMs) have been remarkably successful. LLMs trained with Reinforcement Learning (RL) for reasoning demonstrate strong performance in challenging tasks such as mathematics and coding, even with relatively small model sizes. However, despite these impressive improvements in task accuracy, the assessment of creativity in LLM generations has been largely overlooked in reasoning tasks, in contrast to writing tasks. The lack of research on creativity assessment in reasoning primarily stems from two challenges: (1) the difficulty of defining the range of creativity, and (2) the necessity of human evaluation in the assessment process. To address these challenges, we propose CLAWS, a novel method that defines and classifies mathematical solutions into Typical, Creative, and Hallucinated categories without human evaluation, by leveraging attention weights across prompt sections and output. CLAWS outperforms five existing white-box detection methods—Perplexity, Logit Entropy, Window Entropy, Hidden Score, and Attention Score—on five 7–8B math RL models (DeepSeek, Qwen, Mathstral, OpenMath2, and Oreal). We validate CLAWS on 4,545 math problems collected from 181 math contests (A(J)HSME, AMC, AIME). Our code is available at https://github.com/kkt94/CLAWS.

## 1 Introduction

In recent years, Large Language Models (LLMs) have achieved remarkable success across a wide range of tasks. Among these, the most notable progress has been made in reasoning ability, particularly in mathematical problem solving. Solving math problems requires cognitive processes that go beyond simple calculations, making it an ideal benchmark for assessing how closely AI approximates human intelligence. Recently released frontier LLMs [1, 2, 3] appear to approach human-level intelligence in terms of accuracy on mathematical reasoning tasks.

However, human intelligence is not defined solely by accuracy; it also encompasses diverse aspects such as creativity. Within LLM research, creativity has primarily been explored in writing tasks, often evaluated through the Torrance Test of Creative Writing (TTCW) [4, 5], which is adapted from the Torrance Tests of Creative Thinking (TTCT) [6, 7]. These tests provide a framework for assessing creativity beyond factual consistency between input and output [8] or coherence of generated responses [9]. In contrast, creativity remains largely overlooked in reasoning tasks.

---

*Equal contribution
†Correspondence to: Yong Suk Choi <cys@hanyang.ac.kr>

39th Conference on Neural Information Processing Systems (NeurIPS 2025).

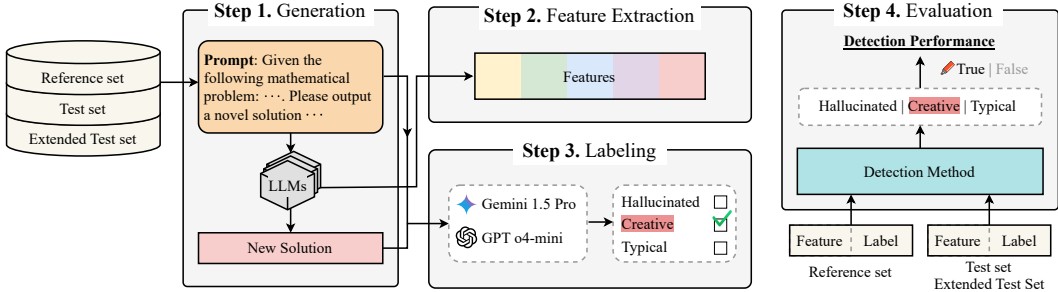

Figure 1: Overview of the proposed framework. **Step 1:** LLM generates a solution. **Step 2:** Features are extracted during generation. **Step 3:** LLM Evaluator assigns labels (Hallucinated / Creative / Typical). **Step 4:** Detection methods are evaluated by comparing predictions with the labels.

Evaluating creativity in reasoning is particularly challenging because it requires human expertise to establish what constitutes a "creative" solution. Assessing mathematical creativity, in particular, demands high-level domain knowledge, making large-scale evaluation costly and difficult to standardize. To overcome these challenges, recent studies on mathematical problem solving [10, 11] have attempted to define creativity and measure the creative problem-solving abilities of LLMs. Interestingly, their results revealed that even LLMs with similar accuracy exhibit substantial differences in creative ability, motivating further research on creativity assessment in reasoning tasks.

If clear criteria for judging creativity were available, it would be possible to detect whether an LLM's response is creative or not. Since the emergence of LLMs, numerous studies have focused on hallucination detection [12, 13]. While detecting and mitigating hallucination is crucial for ensuring factual accuracy, excessive restriction of model generations may inadvertently suppress creativity and lead to repetitive, typical outputs [14]. Thus, identifying creative responses is essential for maximizing the effectiveness and diversity of LLM generations.

Recent advances in prompt engineering have shown that the structure of input prompts significantly affects LLM performance [15, 16]. Building on these findings, we hypothesize that creative generation may depend on which sections of the prompt an LLM attends to—whether it relies more on the given instructions or on its self-generated reasoning. Accordingly, we propose CLAWS, which divides the prompt into sections and quantifies, via attention analysis, the degree to which each section influences generation. This enables detection of Hallucinated, Creative, and Typical solutions based on attention differences across prompt sections and output.

In this study, we present an experimental framework that classifies generated mathematical solutions into Hallucinated, Creative, and Typical categories, and propose CLAWS, a novel white-box detection method that leverages attention over distinct prompt sections and output to perform this classification without requiring human evaluation. CLAWS achieves three-way classification with high efficiency—a capability rarely demonstrated by existing hallucination detection methods. Moreover, it consistently outperforms baseline methods on hallucination detection tasks.

To validate the superior performance of CLAWS, we utilize five Reasoning Language Models (RLMs)—DeepSeek-Math[17], Qwen-Math[18], Mathstral[19], OpenMath2-Llama3.1[20], and Oreal[21]—each with 7–8B parameters and trained with reinforcement learning to enhance reasoning capabilities. We conduct extensive validation using a dataset of 4,545 mathematical problems spanning Algebra, Precalculus, Prealgebra, Number Theory, Geometry, and Counting & Probability.

Our major contributions are summarized as follows:

- We propose a framework for detecting Hallucinated, Creative, and Typical solutions without human evaluation in reasoning tasks using RLMs.

- We introduce CLAWS, a novel white-box method for detecting creativity and hallucination in mathematical reasoning.

- We present a comprehensive evaluation protocol, consisting of five evaluation strategies and four metrics, to assess the features extracted by detection methods.

## 2 Experimental Framework

An overview of the experimental framework is presented in Figure 1. During the generation process, features are extracted from the model's internal representations through the Generator. The generated responses are then labeled by the LLM Evaluator, which determines whether each solution is Hallucinated, Creative, or Typical. Finally, the reference set is utilized to perform detection using the selected methods, enabling each method to classify the responses based on the extracted features.

### 2.1 Problem Formulation

We aim to classify a generated solution $R = f(X)$ into one of three categories — Hallucinated / Creative / Typical Solution — without relying on human evaluation, where $X = G|P|S|I$ is the input prompt to the generative model $f$. As illustrated in Figure 2, input prompt $X$ consists of the following four sections:

- **Guideline** $G$: Describes the criteria for evaluating the difference between two mathematical solutions, providing the model with the concept and standard for identifying Creative solutions.
- **Problem** $P$: The math problem that the model is required to solve.
- **Reference Solutions** $S$: A set of 1 to $n$ typical solutions to the problem, provided to help the model generate a creative solution in contrast to these references.
- **Instruction** $I$: An instruction to create a novel solution that is different from reference solutions for a given problem.

In prior study [11], prompt in Figure 2 was used to induce the generative model to create novel creative solutions, and the creative problem-solving ability of LLMs was successfully presented. Building on this setup, we take a further step by investigating whether the model's internal state can detect Hallucinated / Creative / Typical Solution.

We define a generated response $R$ as a Creative solution if it differs from the provided reference solutions $S$ in a way that satisfies the criteria specified in the guideline $G$. The generative model $f$ receives between 1 and n reference solutions and generates a novel solution accordingly. Each $R$ is then evaluated by a LLM evaluator $E$, using the prompt described in Appendix B.1.

**Criteria for evaluating the difference between two mathematical solutions include:**
i). If the methods used to arrive at the solutions are fundamentally different, such as algebraic manipulation versus geometric reasoning, they can be considered distinct;
ii). Even if the final results are the same, if the intermediate steps or processes involved in reaching those solutions vary significantly, the solutions can be considered different;
iii). If two solutions rely on different assumptions or conditions, they are likely to be distinct;
iv). A solution might generalize to a broader class of problems, while another solution might be specific to certain conditions. In such cases, they are considered distinct;
v). If one solution is significantly simpler or more complex than the other, they can be regarded as essentially different, even if they lead to the same result.

**Given the following mathematical problem:**
What is the largest power of 2 that is a divisor of $13^4 - 11^4$?

**And some typical solutions:**
1. First, we use the difference of squares on
$13^4 - 11^4 = (12^2)^2 - (11^2)^2. \cdots$
2. Just like in the above solution, we use the difference-of-squares factorization, but only once to get
$13^4 - 11^4 = (13^2 - 11^2)(13^2 + 11^2). \cdots$

**Please output a novel solution distinct from the given ones for this math problem.**

Figure 2: Example of an input prompt $X$ used to elicit creative solutions from LLMs. Prompt consists of four sections: Guideline ($G$, yellow), Problem ($P$, green), Reference Solutions ($S$, blue), and Instruction ($I$, purple).

### 2.2 Model

#### 2.2.1 RLM Generator

To generate mathematical problem solutions, we select five RLMs: DeepSeek-Math-7B-RL [17], Qwen2.5-Math-7B-Inst [18], Mathstral-7B [19], OpenMath2-Llama3.1-8B [20], and OREAL-7B [21]. These models are released after 2024 and have 7-8B parameter scales, which enables white-box approach of long length of problem and reference solutions. In addition to the RLM, we also tested general LLMs such as LLaMa3-8B [22], Qwen2.5-14B [23], and DeepSeekV2-16B [24], but finally did not adopt them due to their lack of ability to creatively solve mathematical problems or model sizes.

#### 2.2.2 LLM Evaluator

To evaluate RLM's Generations, we employ two frontier-level LLMs — GPT-o4-mini [25] and Gemini-1.5-Pro [26]— as LLM Evaluators. These two models have shown great performance on

Table 1: Number of samples per class (Hallucinated, Creative, Typical) for each dataset and model. **Ha** denotes Hallucinated solutions, **Cr** Creative solutions, and **Ty** Typical solutions.

| Model | DeepSeek | | | | Mathstral | | | | OpenMath2 | | | | OREAL | | | | Qwen-2.5 | | | |
|---|---|---|---|---|---|---|---|---|---|---|---|---|---|---|---|---|---|---|---|---|
| **Dataset** | Ha | Cr | Ty | Total | Ha | Cr | Ty | Total | Ha | Cr | Ty | Total | Ha | Cr | Ty | Total | Ha | Cr | Ty | Total |
| **REF** | 868 | 206 | 649 | 1723 | 1192 | 175 | 437 | 1804 | 923 | 103 | 785 | 1811 | 1244 | 83 | 379 | 1706 | 631 | 324 | 752 | 1707 |
| **TEST** | 798 | 160 | 456 | 1414 | 961 | 154 | 337 | 1452 | 815 | 97 | 551 | 1463 | 932 | 89 | 369 | 1390 | 578 | 203 | 579 | 1360 |
| **AMC** | 1197 | 530 | 1373 | 3100 | 1679 | 434 | 1049 | 3180 | 1330 | 291 | 1578 | 3199 | 1928 | 237 | 935 | 3100 | 637 | 629 | 1784 | 3050 |
| **AIME** | 772 | 126 | 262 | 1160 | 917 | 68 | 221 | 1206 | 644 | 67 | 501 | 1212 | 911 | 47 | 161 | 1119 | 529 | 159 | 373 | 1061 |
| **A(J)HSME** | 657 | 424 | 763 | 1844 | 945 | 354 | 606 | 1905 | 723 | 248 | 943 | 1914 | 1005 | 161 | 656 | 1822 | 281 | 491 | 1054 | 1826 |

mathematical benchmarks and have already been recognized as human-level Evaluators in prior studies. Based on their evaluations, we assess the solutions according to the following criteria:

- **Hallucinated Solution**: If both evaluator did not judge the generated solution as 'Correctness', it was classified as a Hallucinated Solution.
- **Creative Solution**: Among the generated solutions that both evaluators judged to be 'Correctness', if even one evaluator judged them to be 'Creativity', the solution was classified as Creative Solution. This is a criterion for inclusive acceptance of creativity.
- **Typical Solution**: Among the generated solutions that both evaluators judged as 'Correctness', those that neither evaluator judged as 'Creativity' were classified as Typical Solution.

## 2.3 Dataset

To conduct our study, we adopted publicly available math datasets from CreativeMath [11] and HARP [27]. CreativeMath is a dataset of 400 math problems with solutions, sampled and cleaned from 50 questions from eight math contests: AMC 8, 10, 12, A(J)HSME, AIME, USAJMO, USAMO, and IMO. HARP contains 5,409 problems from A(J)HSME, AMC, AIME, and USA(J)MO. Since there is an overlap of 282 problems between HARP and CreativeMath, We reconstructed HARP by removing duplicate entries and excluding problems that were either too difficult (e.g., proof-based) or whose problems or solutions were excessively long, resulting in a final set of 4,545 problems with solutions.

### 2.3.1 Dataset Construction

We generate solutions using the Generators described in Section 2.2.1, and construct the reference set and test set from the CreativeMath, and the extended test set from the HARP. The reference set serves as a low-resource for detection, the test set is used for validation on the same dataset, and the extended test set is used for validation on an extended dataset. For all Generators, we limit the input token length to 2,048 and the output token length to fewer than 1,024 tokens, based on each Generator's respective tokenizer.

**Reference Set** Reference set serves as the standard for classifying generated solutions into one of three categories. We select 29 problems from the CreativeMath dataset, considering the distribution of difficulty levels. Following the approach of prior work [12], we generate 20 responses for each input prompt using stochastic decoding. For each problem, we include up to $n$ reference solutions, where $n$ is the number of reference solutions available in the dataset. We attempted to ensure sufficient diversity in the reference set by providing a diverse number of reference solutions for each problem. Because each reference solution for problem differs in both length and content, naturally resulted in variation in prompt length. Such diversity contributes to the effectiveness of the reference set as a criterion for classification.

**Test Set** Test set consists of the remaining 371 problems with solutions from CreativeMath. For each problem, three responses are generated using stochastic decoding. The number of reference solutions $n$ provided in the input prompt is limited to a maximum of two.

**Extended Test Set** Extended test set is used to evaluate the generalization performance of the method. We utilize problems and solutions from four math competitions — A(J)HSME, AMC, and AIME — compiled in the HARP. For each problem, one response is generated. The number of reference solutions $n$ included in the prompt is limited to a maximum of two, consistent with the test set.

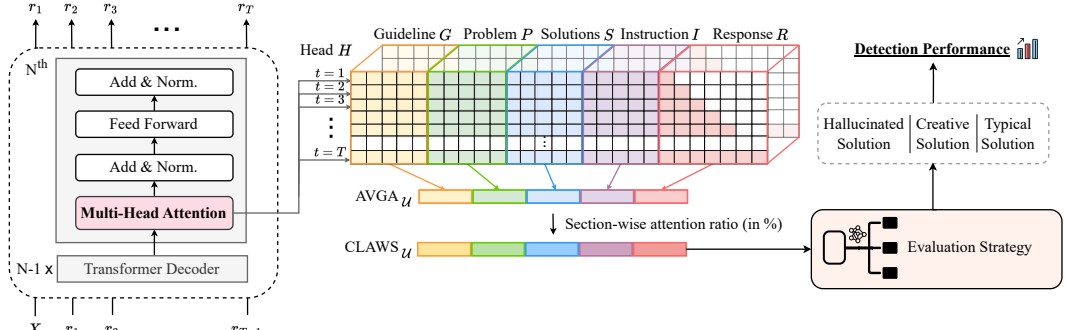

Figure 3: Architecture of CLAWS. All tokens are segmented into five sections: Guideline, Problem, Solutions, Instruction, and Response. The average attention weight for each section ($\text{AVGA}_{\mathcal{U}}$) is computed, normalized to obtain section-wise attention ratios ($\text{CLAWS}_{\mathcal{U}}$), and used as features for detecting Hallucinated, Creative, or Typical solutions.

## 3 CLAWS:Creativity detection for LLM-generated solutions using Attention Window of Sections

In Figure 3, we input the prompt $X = G|P|S|I$ into the RLM $f$ and leverage the decoder attention weights during generation to classify the class of the generated response $R = f(X)$. As described in Section 2.1, $X$ consists of four sections — Guideline ($G$), Problem ($P$), reference Solutions ($S$), and Instruction ($I$). we additionally include the generated Response section ($R$), thereby dividing tokens into five distinct semantic segments. We hypothesize that the class of $R$ is influenced by which sections the model attends to most during generation. Based on this hypothesis, we propose CLAWS, a method that leverages the average attention weights for each section windows. Let $A_{t,h}^{(L)}$ denote the attention weights at decoding time step $t$ from head $h$ in last layer $L$:

$$A_{t,h}^{(L)} = [a_{1,h}^{(L)}\ a_{2,h}^{(L)} \cdots a_{k,h}^{(L)} \cdots a_{k+t,h}^{(L)}], \text{ where } k = len(X)$$

To construct the complete attention weight matrix $A_h^{(L)}$, we stack the attention vectors over all time steps $t = 1 \cdots T$. Since one output token is generated at each time step, the length of the attention target increases over time. To ensure consistent dimensionality, each $A_{t,h}^{(L)}$ is padded to a fixed length of $len(X) + T$.

$$A_h^{(L)} = \begin{bmatrix} A_{1,h}^{(L)} & A_{2,h}^{(L)} & \cdots & A_{T,h}^{(L)} \end{bmatrix}^T \in \mathbb{R}^{T \times (len(X)+T)}$$

We then compute the average attention weight for each section $\mathcal{U} \in G, P, S, I, R$ by summing the attention weights over all tokens belonging to that section, and averaging over all heads $H$, time steps $T$, and section-specific token positions $\mathcal{I}_{\mathcal{U}}$:

$$\text{AVGA}_{\mathcal{U}} = \frac{1}{H \cdot T \cdot |\mathcal{I}_{\mathcal{U}}|} \sum_{h=1}^{H} \sum_{t=1}^{T} \sum_{i \in \mathcal{I}_{\mathcal{U}}} A_h^{(L)}[t, i], \text{ for } \mathcal{U} = \{G, P, S, I, R\}$$

Finally, to quantify the proportion of attention allocated to each section, we normalize the average attention weights AVGA into ratios and use them as input features for the evaluation model:

$$\text{CLAWS}_{\mathcal{U}} = \frac{\text{AVGA}_{\mathcal{U}}}{\sum_{\mathcal{U}' \in \{G,P,S,I,R\}} \text{AVGA}_{\mathcal{U}'}}$$

CLAWS extracts only the attention layer in the single response generation process and just performs sum and average operations, enabling effective detection without additional calls or operations. This

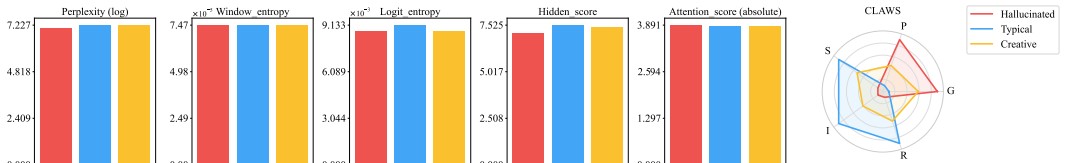

Figure 4: Visualization of class-wise average scores for each method, computed on the reference set generated using Qwen2.5-math-7B-inst. To enhance visual clarity, the normalization range is clipped between 0.1 and 0.9 for CLAWS. Visualizations for all models are presented in Figure 9.

is an efficient method with a similar level of computational overhead to methods that only use existing hidden state or attention layers.

# 4 Evaluation

## 4.1 Baseline

In previous research, both black-box, including SelfCheckGPT [12], and white-box approaches that use uncertainty-based methods [28] have been extensively studied to detect hallucinations in LLM generations. In addition, there are methods to mitigate hallucinated in LLMs through majority voting by generating multiple generations, such as Self-Consistency [29] or INSIDE [30]. However, these methods have limitations, as they depend on external models and require multiple responses per input, resulting in substantial computational cost. Therefore, these methods are not suitable as baselines for this study. In light of these limitations, we employ five white-box hallucination detection methods as baselines in this study, without relying on external models or using majority voting, to distinguish between Hallucinated, Creative, and Typical solutions. The five methods can be broadly categorized into output token uncertainty quantification and eigenvalue analysis of internal LLM representations. These approaches have been widely used in prior research, and more recently, LLM-Check [31] has further consolidated these methods.

**Perplexity** Perplexity is a measure of the confidence in the model's generation of the the response $\mathbf{x} = (x_{n+1} \cdots x_m)$, which is generated by the model $f$ given an input prompt $\mathbf{x}_p$. It is calculated based on the log-likelihood of each output token $x_i$, where $m$ denotes the length of the generated response and $n$ denotes the start index of the response being evaluated and is defined as follows:

$$\text{Perplexity}(\mathbf{x}) = \exp\left(-\frac{1}{m-n+1}\sum_{i=n}^{m}\log p_f\big(x_i \mid \mathbf{x}_p \oplus \mathbf{x}_{<i}\big)\right) \tag{1}$$

**Logit Entropy** Logit Entropy represents the uncertainty in the generation by measuring the entropy of the probability distribution over the top-$k$ tokens at each token position, based on the logits:

$$\text{LogitEntropy}(\mathbf{x}, k) = -\frac{1}{m-n+1}\sum_{i=n}^{m}\sum_{j=1}^{k} p_f\big(x_i^j \mid \mathbf{x}_p \oplus \mathbf{x}_{<i}\big)\log p_f\big(x_i^j \mid \mathbf{x}_p \oplus \mathbf{x}_{<i}\big). \tag{2}$$

**Window Logit Entropy** Since Logit Entropy calculates over the entire response, there is a problem that the entropy value is diluted and hallucinations cannot be detected when hallucinations is short. In order to address the problem, Window Logit Entropy is defined as the calculation of the logit entropy over sliding windows of size $k$ and selecting the maximum value among them, as follows:

$$\text{WindowLogitEntropy}(\mathbf{x}, k, w) = \max_{s \in \{1, \ldots, m-w+1\}}\left\{\frac{1}{w}\sum_{i=s}^{s+w-1}\text{LogitEnt}(x_i, k)\right\}. \tag{3}$$

**Hidden Score** Hidden Score is a measure of the variance in representation diversity, computed by performing an eigen-decomposition on the hidden state matrix $\mathbf{H} \in \mathbb{R}^{d \times m}$, which consists of $m$

tokens represented embedding in $d$ dimensional space, and then calculating the mean of the logarithm of eigenvalues. So, Hidden Score is defined as the mean log-determinant of $\Sigma^2$ where $\sigma_i$ represent the singular values of $\mathbf{H}$:

$$\text{HiddenScore} = \frac{1}{m} \log \det(\Sigma^2) = \frac{1}{m} \sum_{i=1}^{m} \log \sigma_i, \quad \text{where } \Sigma^2 = \mathbf{H}^\top \mathbf{H}. \tag{4}$$

**Attention Score** Attention Score quantifies self-attention by analyzing the diagonal entries of the self-attention kernel matrix:

$$\text{AttentionScore} = \frac{1}{m} \sum_{j=1}^{m} \log\big((\text{Ker}_i)^{jj}\big). \tag{5}$$

$Ker_i^{jj}$ denotes the $j$-th diagonal entry of the self-attention kernel matrix from $i$-th attention head.

## 4.2 Evaluation Strategy

The standard evaluation strategy for the five baselines is to determine a threshold that yields the optimal detection performance. However, the limitations of threshold-based approaches have been consistently noted, and several studies have explored how to better leverage internal model features for detection. In this study, we evaluate the performance of CLAWS and the five baselines using five distinct evaluation strategies. Detailed hyperparameters and experimental settings are provided in Appendix B.3.

**Threshold** A threshold that yields the best performance among the values computed by each detection method is determined. This strategy applies only to the five baselines that use a single scalar value as a feature.

**Prototypes** A prototype-based classification approach is employed using the reference set [32]. The Euclidean distance between each class's center embedding and a given sample is computed to predict the nearest class. This strategy applies only to CLAWS, which outputs multi-dimensional feature vectors.

**XGBOOST** A tree-based classification algorithm based on Gradient Boosting Decision Trees (GBDT) is employed. XGBoost [33] serves as an advanced implementation of this approach.

**MLP** A conventional trainable classifier is employed to classify the features extracted by each method. The classifier is implemented as a Multi-Layer Perceptron (MLP).

**TabM** An ensemble-based classification method is employed to mitigate the large variance of MLPs. TabM [34] trains multiple MLPs and aggregates their predictions to generate the final prediction.

## 4.3 Evaluation Metrics

For a fair comparison, we employ four evaluation metrics: weighted F1 score ($F1_w$), macro F1 score ($F1_m$), Area Under the ROC Curve (AUROC), and macro Average Precision ($AP_m$). Given the limited number of samples and the intrinsic difficulty of classifying the Creative class—which lies between the Typical and Hallucinated classes—it is essential to adopt metrics that capture the model's ability to recognize rare and challenging categories. Furthermore, in cases where not all three classes were predicted, we separately marked instances where a model achieved an artificially high score by predicting only the majority class, indicated by gray highlighting in the tables.

## 5 Result

In Table 2, five baselines were evaluated using thresholds, while CLAWS was assessed with a prototype strategy; CLAWS outperformed all models on the test set across all four metrics. The full result table is presented in the Appendix C, using all strategies and metrics. In particular, CLAWS achieved superior performance in $F1_m$ and $AP_m$, metrics that measure the macro-average by giving equal weight to all classes. In contrast, none of the five baselines performed well on all models.

Table 2: Results for creativity detection. The evaluation strategies are Threshold (for PPL: Perplexity, WE: Window Entropy, LE: Logit Entropy, HS: Hidden Score, AS: Attention Score) and Prototype (for CLAWS). Bold values indicate the best performance, underlined values denote the second best, and gray-shaded cells correspond to cases where the model detected only two out of the three classes.

| Dataset | | TEST | | | | AMC | | | | AIME | | | | A(J)HSME | | | |
|---|---|---|---|---|---|---|---|---|---|---|---|---|---|---|---|---|---|
| Model | Method | $F1_w$ | $F1_m$ | $AP_m$ | AUROC | $F1_w$ | $F1_m$ | $AP_m$ | AUROC | $F1_w$ | $F1_m$ | $AP_m$ | AUROC | $F1_w$ | $F1_m$ | $AP_m$ | AUROC |
| DeepSeek | PPL | 48.09 | 35.77 | 37.07 | 56.49 | 44.56 | 35.63 | 36.28 | 55.12 | 55.93 | 36.70 | 36.59 | 56.63 | 42.34 | 36.52 | 37.49 | 56.82 |
| | WE | 18.59 | 23.20 | 35.89 | 53.89 | 27.52 | 26.03 | 34.67 | 52.26 | 9.07 | 16.36 | 33.67 | 50.31 | 28.72 | 27.70 | 34.44 | 51.92 |
| | LE | 40.56 | 33.21 | 34.00 | 50.90 | 35.69 | 32.96 | 33.42 | 50.31 | 28.99 | 25.28 | 33.27 | 48.89 | 34.89 | 33.46 | 33.45 | 50.18 |
| | HS | 29.56 | 25.18 | 32.40 | 45.03 | 38.44 | 32.96 | 33.60 | 50.22 | 38.30 | 29.65 | 33.71 | 50.67 | 38.61 | 35.80 | 34.54 | 52.40 |
| | AS | 33.95 | 24.99 | 30.98 | 42.80 | 33.51 | 29.18 | 33.43 | 50.19 | 43.92 | 32.89 | 33.58 | 50.97 | 28.63 | 26.83 | 33.19 | 49.70 |
| | CLAWS | 58.66 | 46.01 | 41.17 | 62.09 | 46.71 | 40.99 | 37.16 | 56.40 | 56.90 | 38.12 | 35.38 | 54.47 | 38.82 | 37.64 | 36.25 | 54.40 |
| Mathstral | PPL | 42.45 | 25.94 | 31.37 | 43.26 | 36.50 | 25.21 | 31.81 | 45.89 | 56.90 | 29.97 | 32.76 | 47.49 | 34.79 | 25.58 | 32.58 | 48.15 |
| | WE | 46.19 | 28.89 | 32.58 | 46.68 | 40.71 | 30.02 | 32.73 | 48.44 | 52.20 | 30.20 | 32.91 | 48.33 | 40.34 | 31.79 | 33.86 | 51.05 |
| | LE | 41.62 | 28.17 | 32.11 | 45.66 | 35.20 | 29.55 | 32.05 | 46.93 | 44.77 | 28.56 | 33.47 | 50.50 | 35.46 | 30.56 | 32.40 | 47.77 |
| | HS | 49.86 | 26.53 | 32.49 | 47.07 | 37.37 | 23.46 | 33.33 | 49.97 | 65.96 | 31.13 | 33.46 | 50.23 | 33.42 | 22.65 | 33.42 | 50.14 |
| | AS | 38.41 | 24.50 | 31.23 | 42.22 | 36.92 | 27.53 | 32.02 | 46.69 | 57.35 | 31.95 | 33.51 | 49.82 | 35.26 | 27.57 | 32.41 | 47.60 |
| | CLAWS | 63.20 | 46.05 | 41.75 | 63.70 | 51.47 | 41.45 | 37.89 | 57.69 | 65.25 | 36.05 | 34.43 | 52.73 | 49.13 | 42.29 | 38.20 | 58.18 |
| OpenMath2 | PPL | 36.47 | 27.52 | 32.72 | 47.30 | 41.10 | 31.45 | 33.12 | 49.24 | 40.44 | 30.49 | 32.18 | 47.57 | 39.22 | 30.05 | 33.13 | 48.56 |
| | WE | 40.89 | 32.14 | 33.84 | 50.50 | 43.44 | 34.48 | 33.93 | 51.19 | 40.55 | 31.17 | 33.37 | 50.00 | 42.45 | 34.16 | 33.99 | 51.08 |
| | LE | 47.48 | 35.96 | 35.15 | 53.15 | 43.17 | 36.18 | 34.28 | 52.62 | 41.82 | 33.10 | 34.32 | 52.92 | 42.38 | 37.55 | 34.70 | 53.28 |
| | HS | 30.48 | 23.20 | 30.77 | 41.57 | 33.02 | 26.78 | 31.17 | 44.52 | 40.45 | 32.09 | 32.63 | 49.34 | 31.62 | 26.55 | 31.37 | 44.93 |
| | AS | 33.20 | 24.48 | 30.65 | 42.17 | 32.84 | 27.77 | 31.89 | 46.75 | 40.42 | 30.96 | 48.59 | 32.59 | 31.03 | 27.53 | 32.09 | 47.04 |
| | CLAWS | 60.86 | 44.27 | 40.77 | 60.66 | 54.32 | 42.12 | 38.53 | 58.06 | 49.35 | 34.41 | 35.35 | 52.00 | 50.88 | 41.36 | 37.73 | 57.22 |
| OREAL | PPL | 46.52 | 27.81 | 31.78 | 45.60 | 41.68 | 26.38 | 31.25 | 44.27 | 55.96 | 28.11 | 32.64 | 47.36 | 36.90 | 24.83 | 31.03 | 43.62 |
| | WE | 49.57 | 27.39 | 32.80 | 48.26 | 44.87 | 27.32 | 32.87 | 48.82 | 66.65 | 32.63 | 33.48 | 51.28 | 36.37 | 24.79 | 32.79 | 48.44 |
| | LE | 55.39 | 36.15 | 34.46 | 53.11 | 49.80 | 35.95 | 34.43 | 53.30 | 63.53 | 33.86 | 34.02 | 53.06 | 41.06 | 31.86 | 33.29 | 50.47 |
| | HS | 51.95 | 29.46 | 32.60 | 47.90 | 48.58 | 28.28 | 33.20 | 49.60 | 68.10 | 31.63 | 33.30 | 49.22 | 41.65 | 28.36 | 33.12 | 49.62 |
| | AS | 45.56 | 28.24 | 31.83 | 45.56 | 47.92 | 29.11 | 32.70 | 48.25 | 65.19 | 32.74 | 33.20 | 49.56 | 40.18 | 26.41 | 32.74 | 48.48 |
| | CLAWS | 54.19 | 40.18 | 38.15 | 59.46 | 43.83 | 34.77 | 35.57 | 54.78 | 59.95 | 32.74 | 33.81 | 51.55 | 35.70 | 31.93 | 35.51 | 54.41 |
| Qwen-2.5 | PPL | 25.66 | 23.30 | 31.76 | 42.62 | 26.40 | 21.39 | 31.71 | 43.31 | 28.29 | 25.29 | 32.00 | 44.52 | 24.88 | 20.34 | 32.09 | 44.79 |
| | WE | 30.79 | 29.40 | 34.71 | 52.50 | 22.04 | 26.04 | 33.80 | 51.15 | 33.23 | 29.08 | 33.12 | 49.24 | 20.50 | 23.61 | 33.12 | 49.51 |
| | LE | 50.81 | 45.29 | 39.50 | 59.80 | 45.86 | 40.18 | 36.15 | 55.81 | 43.20 | 39.01 | 36.40 | 55.83 | 45.70 | 38.64 | 35.23 | 54.09 |
| | HS | 30.67 | 28.31 | 36.25 | 54.53 | 47.57 | 31.98 | 34.81 | 52.98 | 20.37 | 24.17 | 34.57 | 52.83 | 48.52 | 32.54 | 34.78 | 52.77 |
| | AS | 30.75 | 26.96 | 32.05 | 45.53 | 38.61 | 31.55 | 32.93 | 48.78 | 37.32 | 33.71 | 33.92 | 51.42 | 33.72 | 28.55 | 32.86 | 48.35 |
| | CLAWS | 50.35 | 43.37 | 39.88 | 59.32 | 52.77 | 41.39 | 37.45 | 57.59 | 39.05 | 32.31 | 33.08 | 49.31 | 47.90 | 36.04 | 35.86 | 54.94 |

However, CLAWS demonstrated superiority in $F1_m$ and $AP_m$, which reflect the macro-average, and showed even more pronounced performance in $F1_w$, which assigns more weight to larger sample classes. This finding suggests that CLAWS not only effectively detects Creative solutions but also maintains robust performance in classifying Typical and Hallucinated solutions.

As shown in Figure 4 and Figure 9, the reason for CLAWS superior performance is clearly presented. The visualization illustrates the mean per method for each class based on 20 generations for the same prompt in the reference set. In the reference set, which is the basis for detection, the five baselines record averages that do not differ significantly by class. In contrast, CLAWS effectively distinguishes Creative solutions as distributed between Hallucinated and Typical solutions.

In Figure 4, the Hallucinated solutions are relatively attention in the Guideline and Problem sections, while the Typical solutions are attention in the reference Solution, Instruction, and Response sections. Figure 4 and 9 show that hallucination focus on the Guideline section in all models. This suggests that hallucinations may be caused by over-focusing on a part of the input prompt.

A similar result is observed in extended test set. CLAWS performed well in most models, except for the OREAL. This is because the OREAL has a much higher rate of generating Hallucinated solutions than others, as shown in Table 1. Therefore, it was challenging to produce a prototype well, which consequently made the classification difficult. As shown in Table 20, however, it achieves significantly higher performance with evaluation strategies as MLP, XGBoost, and TabM.

Table 3: Results for hallucination detection. Threshold (PPL, WE, LE, HS, AS) and Prototype (CLAWS) are used as evaluation strategies. Bold and underlined values indicate the best and second-best performance. Gray-shaded cells indicate cases where the model predicted only a single class.

| Dataset | | TEST | | | | AMC | | | | AIME | | | | A(J)HSME | | | |
|---|---|---|---|---|---|---|---|---|---|---|---|---|---|---|---|---|---|
| Model | Method | $F1_w$ | $F1_m$ | $AP_m$ | AUROC | $F1_w$ | $F1_m$ | $AP_m$ | AUROC | $F1_w$ | $F1_m$ | $AP_m$ | AUROC | $F1_w$ | $F1_m$ | $AP_m$ | AUROC |
| DeepSeek | PPL | 37.63 | 40.32 | 45.13 | 53.06 | 52.18 | 45.38 | 62.33 | 51.95 | 39.75 | 42.77 | 36.16 | 55.60 | 55.31 | 46.20 | 65.42 | 52.25 |
| | WE | 26.44 | 30.34 | 43.56 | 50.00 | 46.70 | 38.04 | 61.39 | 50.00 | 16.77 | 25.06 | 33.45 | 50.00 | 50.42 | 39.16 | 64.37 | 50.00 |
| | LE | 27.66 | 31.46 | 43.81 | 50.50 | 46.82 | 38.19 | 61.40 | 50.03 | 17.22 | 25.35 | 33.36 | 49.81 | 50.37 | 39.12 | 64.33 | 49.92 |
| | HS | 26.41 | 30.31 | 43.52 | 49.92 | 47.11 | 38.56 | 61.48 | 50.20 | 17.64 | 25.74 | 33.54 | 50.19 | 50.61 | 39.51 | 64.34 | 49.93 |
| | AS | 26.60 | 30.45 | 43.43 | 49.72 | 46.68 | 38.02 | 61.37 | 49.97 | 16.77 | 25.06 | 33.45 | 50.00 | 50.59 | 39.43 | 64.38 | 50.03 |
| | CLAWS | **67.46** | **67.24** | **55.73** | **67.78** | **61.77** | **59.79** | **66.64** | **59.84** | **61.95** | **58.17** | **38.45** | **58.68** | **64.93** | **61.67** | **70.38** | **61.62** |
| Mathstral | PPL | 17.09 | 25.27 | 33.82 | 50.00 | 29.62 | 31.76 | 46.58 | 49.90 | 9.44 | 19.45 | 23.98 | 50.05 | 33.84 | 33.58 | 50.37 | 49.95 |
| | WE | 17.09 | 25.27 | 33.82 | 50.00 | 29.71 | 31.84 | 46.62 | 49.96 | 9.61 | 19.57 | 24.00 | 50.11 | 33.89 | 33.63 | 50.42 | 50.05 |
| | LE | 17.36 | 25.47 | 33.82 | 50.00 | 29.65 | 31.79 | 46.62 | 49.97 | 10.12 | 19.94 | 24.06 | 50.27 | 34.24 | 33.98 | 50.50 | 50.21 |
| | HS | 17.03 | 25.08 | 33.48 | 49.24 | 30.11 | 32.18 | 46.36 | 49.44 | 9.26 | 19.33 | 23.96 | 50.00 | 33.96 | 33.70 | 50.39 | 50.00 |
| | AS | 17.09 | 25.27 | 33.82 | 50.00 | 29.66 | 31.80 | 46.64 | 50.00 | 9.26 | 19.33 | 23.96 | 50.00 | 33.77 | 33.51 | 50.39 | 50.00 |
| | CLAWS | **72.99** | **69.59** | **49.42** | **69.30** | **63.97** | **63.90** | **55.69** | **64.04** | **65.62** | **50.26** | **24.19** | **50.59** | **61.05** | **61.04** | **57.10** | **61.04** |
| OpenMath2 | PPL | 31.78 | 34.68 | 44.16 | 49.74 | 45.53 | 40.17 | 58.19 | 49.51 | 34.68 | 36.15 | 45.52 | 47.20 | 49.49 | 41.63 | 61.65 | 48.76 |
| | WE | 27.19 | 30.70 | 44.29 | 50.00 | 43.09 | 36.88 | 58.42 | 50.00 | 29.91 | 31.91 | 46.86 | 50.00 | 47.74 | 38.36 | 62.23 | 50.00 |
| | LE | 28.14 | 31.55 | 44.39 | 50.20 | 43.18 | 36.99 | 58.41 | 49.97 | 30.38 | 32.35 | 46.89 | 50.06 | 48.09 | 38.81 | 62.32 | 50.21 |
| | HS | 27.55 | 31.01 | 44.27 | 49.95 | 43.40 | 37.27 | 58.43 | 50.01 | 31.30 | 33.21 | 46.95 | 50.17 | 47.68 | 38.37 | 62.12 | 49.78 |
| | AS | 27.46 | 30.94 | 44.32 | 50.05 | 43.30 | 37.16 | 58.39 | 49.92 | 30.09 | 32.08 | 46.90 | 50.08 | 48.14 | 38.90 | 62.30 | 50.15 |
| | CLAWS | **64.91** | **64.70** | **54.19** | **65.05** | **62.53** | **61.65** | **65.19** | **61.82** | **58.10** | **57.50** | **52.32** | **58.47** | **63.88** | **61.81** | **68.70** | **61.96** |
| OREAL | PPL | 16.96 | 25.09 | 32.61 | 49.23 | 21.37 | 27.87 | 37.65 | 49.67 | 6.79 | 16.08 | 18.33 | 49.13 | 28.09 | 31.19 | 44.51 | 49.32 |
| | WE | 23.20 | 29.92 | 33.39 | 50.99 | 22.04 | 28.48 | 37.93 | 50.26 | 10.95 | 18.75 | 18.58 | 49.96 | 28.10 | 31.27 | 44.88 | 50.09 |
| | LE | 20.75 | 27.91 | 32.84 | 49.75 | 23.17 | 29.34 | 37.87 | 50.14 | 10.03 | 18.33 | 18.77 | 50.60 | 30.50 | 33.45 | 45.22 | 50.75 |
| | HS | 16.33 | 24.78 | 32.95 | 50.00 | 20.69 | 27.37 | 37.73 | 49.83 | 6.01 | 15.80 | 18.60 | 50.05 | 27.74 | 30.93 | 44.81 | 49.94 |
| | AS | 16.47 | 24.78 | 32.71 | 49.45 | 20.99 | 27.59 | 37.71 | 49.79 | 6.17 | 15.85 | 18.55 | 49.87 | 28.04 | 31.11 | 44.32 | 48.94 |
| | CLAWS | **58.13** | **56.36** | **38.36** | **59.87** | **53.10** | **53.06** | **41.17** | **56.34** | **64.02** | **49.22** | **18.98** | **51.22** | **53.96** | **54.41** | **48.36** | **56.42** |
| Qwen-2.5 | PPL | 41.30 | 35.91 | 56.88 | 48.72 | 41.30 | 35.91 | 56.88 | 48.72 | 33.74 | 33.64 | 50.05 | 49.81 | 76.90 | 46.45 | 84.45 | 49.38 |
| | WE | 41.98 | 36.51 | 57.50 | 50.00 | 41.98 | 36.51 | 57.50 | 50.00 | 33.70 | 33.61 | 50.19 | 50.09 | **77.69** | 46.20 | 84.66 | 50.18 |
| | LE | 47.87 | 43.32 | 58.86 | 52.72 | 47.87 | 43.32 | 58.86 | 52.72 | 38.32 | 38.24 | 51.16 | 51.99 | 77.58 | 46.13 | 84.62 | 50.05 |
| | HS | 42.12 | 36.67 | 57.51 | 50.02 | 42.12 | 36.67 | 57.51 | 50.02 | 33.49 | 33.40 | 50.14 | 50.00 | 77.53 | 45.82 | 84.60 | 49.97 |
| | AS | 41.92 | 36.45 | 57.44 | 49.87 | 41.92 | 36.45 | 57.44 | 49.87 | 33.49 | 33.40 | 50.14 | 50.00 | 77.44 | 46.04 | 84.58 | 49.87 |
| | CLAWS | **54.67** | **53.30** | **59.21** | **53.35** | **72.13** | **55.12** | **80.75** | **54.82** | **47.08** | **47.10** | 49.11 | 47.82 | 74.68 | **52.33** | **85.25** | **52.44** |

# 6 Analysis

For the Hallucination Detection, we used Typical and Creative classes as Non-hallucinated classes in the dataset presented in Table 1. As shown in Table 3, even though the five baselines are designed to detect hallucinations, CLAWS shows overwhelmingly superior performance. CLAWS showed overwhelming performance not only on the test set but also on three extended test sets, and in particular, on AIME, which contains many difficult math problems, it showed clear discrimination.

When comparing the models, Qwen, which exhibited the most balanced distribution between hallucinated and non-hallucinated samples, consistently achieved high detection performance across all methods. In contrast, all five baselines performed poorly on Mathstral and Oreal, both of which produced approximately 1,200 Hallucinated solutions in the reference set, corresponding to a hallucination rate of nearly 70%. DeepSeek and OpenMath2 displayed distinct patterns depending on the dataset difficulty: in AMC and A(J)HSME, which are relatively easy datasets, the baselines achieved moderate performance, whereas in AIME, their performance dropped substantially.

To overcome these problems, We construct a balanced dataset for creativity detection. For each reference set, a balanced dataset is constructed by including all samples from the minority class (Creative) and randomly sampling an equal number of instances from each of the two majority classes. For example, in the reference set of DeepSeek, there are 206 samples in the Creative class; accordingly, 206 samples are randomly selected from each of the Hallucinated and Typical classes to ensure balance. As shown in Table 16, CLAWS also achieved the best overall performance. In

Table 4: Impact of each prompt section. Performance comparison of the DeepSeek-Math-7B-RL using the Prototype strategy through an ablation study, where each section is selectively removed.

| Dataset | Metric | w/o $G$ | w/o $P$ | w/o $S$ | w/o $I$ | w/o $R$ | Full |
|---|---|---|---|---|---|---|---|
| TEST | $F1_w$ | 58.59 | 58.81 | 50.01 | 54.68 | 58.39 | **58.66** |
| | $F1_m$ | 46.01 | **46.35** | 39.45 | 43.87 | 45.97 | 46.01 |
| | $AP_m$ | 41.13 | 41.04 | 38.46 | 40.29 | 40.60 | **41.17** |
| | AUROC | 62.12 | **62.21** | 59.15 | 61.64 | 61.58 | 62.09 |
| AMC | $F1_w$ | 46.66 | 46.68 | 46.48 | 45.52 | 46.49 | **46.71** |
| | $F1_m$ | 39.59 | 39.33 | 39.59 | 38.16 | 40.20 | **40.99** |
| | $AP_m$ | 37.15 | 37.08 | 36.91 | 36.70 | 37.14 | **37.16** |
| | AUROC | 56.11 | 56.23 | 56.00 | 55.80 | 56.52 | **56.40** |
| AIME | $F1_w$ | 55.88 | 53.33 | 54.92 | 51.37 | 54.01 | **56.90** |
| | $F1_m$ | 37.35 | 35.52 | 36.82 | 34.80 | 34.84 | **38.12** |
| | $AP_m$ | 35.25 | 35.20 | 34.98 | 35.03 | 35.15 | **35.38** |
| | AUROC | 54.08 | 53.38 | 53.51 | 52.47 | 52.67 | **54.47** |
| A(J)HSME | $F1_w$ | 37.85 | 39.92 | **40.49** | 40.10 | 34.25 | 38.82 |
| | $F1_m$ | 37.46 | 35.76 | 37.65 | 36.02 | 34.11 | **37.64** |
| | $AP_m$ | 36.16 | 35.51 | 35.78 | 35.69 | 35.42 | **36.25** |
| | AUROC | **55.05** | 53.88 | 54.33 | 54.18 | 53.08 | 54.40 |

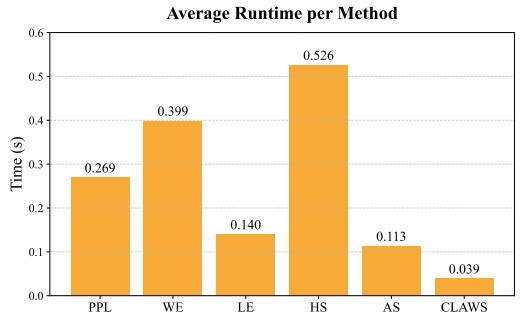

Figure 5: Average runtime for computing input features of each method — PPL, WE, LE, HS, AS, and CLAWS. The Response ($R$) generation time, identical across methods, is excluded.

particular, it performed best under the MLP strategy and exhibited the least performance degradation compared to other methods when trained on a small amount of data.

Furthermore, we analyzed the effect of each prompt section in CLAWS. As shown in Table 4, CLAWS achieved the best overall performance across most datasets and metrics when all five sections were utilized. Moreover, as illustrated in Figures 4 and 9, the degree to which each section influenced generation varied depending on the model. This observation suggests that leveraging all five sections provides the most stable and robust performance regardless of the dataset or model.

Lastly, We compare the runtime efficiency of our proposed method, CLAWS, with five baseline methods. Unlike other approaches, CLAWS does not require any additional mechanisms after the generation phase. Moreover, during generation, it performs only simple operations such as taking the mean or sum of attention weights, without relying on complex computations. As shown in Figure 5, CLAWS demonstrates the highest efficiency among all baselines. Specifically, among entropy-based methods (PPL, WE, and LE), WE, which computes entropy separately for each window, shows the poorest efficiency, whereas LE, which calculates a simple logit entropy, achieves the best efficiency in this group. Among layer-level methods, HS, which involves computing a transposed matrix and singular values, exhibits the lowest efficiency. In contrast, AS, which utilizes only the diagonal components of the self-attention kernel matrix, shows the best efficiency except for CLAWS. In conclusion, CLAWS not only achieves the highest creativity and hallucination detection performance but also exhibits the greatest computational efficiency, demonstrating superiority in all respects.

## 7 Conclusion

In this work, we presented a challenging study on creativity detection. To address this problem, we systematically investigated multiple key components, including the definition of creativity, the design of an experimental framework, the proposal of a novel detection method, and the analysis of evaluation strategies for extracted features. These efforts provide a solid foundation for future research in this emerging area. Moreover, extending hallucination detection toward creativity detection broadens the scope of LLM research beyond reasoning improvement, introducing a new perspective on model evaluation and generation analysis. We anticipate that the proposed experimental framework and our method, CLAWS, will serve as a foundation for future research on reliable creativity detection and the improvement of LLM generation quality and diversity.

## Acknowledgments and Disclosure of Funding

This work was supported by the Institute of Information and communications Technology Planning and evaluation (IITP) grant (No.RS-2025-25422680, No. RS-2020-II201373), and the National Research Foundation of Korea (NRF) grant (No. RS-2025-00520618) funded by the Korean Government (MSIT).

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

# Appendix

## A  Limitations

We have successfully experimented with many RLMs, but have not been able to experiment with general LLMs of similar size because they do not have sufficient creative solution-generating capabilities. Due to the nature of the White-Box approach, using large-sized models (over 20B) with sufficient performance requires a lot of resources. Lastly, we defined 'Creativity' based on solving mathematical problems, and expanding it to a various tasks will be our future work.

## B  Implementation Details

### B.1  LLM Evaluator Details

We use an LLM-based evaluator $E$ to classify each generated response $R$ into three categories — Hallucinated Solution, Typical Solution, or Creative Solution — following the evaluation protocol introduced in [11]. As described in Section 2.2.2 and Figure 1, the evaluation process consists of two stages. First, we assess whether the generated response $R$ is mathematically correct. For example, As shown in Figure 6, the evaluator is given two reference solutions and asked to determine whether $R$ is a valid solution to the given problem.

If both evaluators agree that $R$ is correct, the response proceeds to the second stage, where it is further classified as either a Typical Solution or a Creative Solution. As shown in Figure 7, this decision is made by comparing $R$ against the reference solutions $S$ provided in the original input prompt $X$, and determining whether it satisfies the criteria outlined in the guideline $G$.

Unlike [11], we used two LLM evaluators. In particular, instead of using GPT-4o as one of the evaluators, we adopted o4-mini, which demonstrates superior performance in mathematical evaluation. The LLM evaluators used in our study are listed below:

- **Gemini-1.5-Pro**: `models/gemini-1.5-pro-002`
- **GPT-o4-mini**: `o4-mini-2025-04-16`

### B.1.1  Justification of LLM-based Labeling without Human Evaluation

In this study, we adopted LLM-based labeling instead of human evaluation for dataset construction. The justification is as follows.

**Practical constraints**  The dataset used in our study consisted of mathematically challenging problems, where evaluating the creativity of generated solutions would require human experts with substantial mathematical expertise. However, as shown in Table 1, conducting human evaluation on all 46,528 data instances would have required excessive time and cost, making it practically infeasible.

**Support from prior work**  Recent studies have validated the use of LLMs for creativity assessment [11, 35]. Following these works, we adopted their definitions and evaluation criteria to construct our dataset without relying on human evaluation.

**Indirect reflection of human creativity**  As mentioned earlier, creativity evaluation required a reference solution $S$. Importantly, $S$ was always taken from human-written solutions and never generated by an LLM. Thus, although no direct human evaluation was performed, the diversity and creativity embedded in human-written solutions were indirectly reflected in the evaluation.

**Ensuring reliability**  To enhance the reliability of LLM-based evaluation, we designed the LLM Evaluator not only to output labels but also to provide explicit reasoning. Specifically, we appended the instruction "Additionally, explain the reason for your judgment." at the end of the evaluation prompts (see Figure 6 and 7). Figure 8 shows actual evaluation cases, confirming that the judgments of the LLM Evaluator were consistent and reasonable.

**Given the following mathematical problem:**
What is the largest power of 2 that is a divisor of \(13^4-11^4\)?

**Reference solutions:**
**Solution 1:**
The factors of 36 are 1, 2, 3, 4, 6, 9, 12, 18, and 36. The multiples of 4 up to 36 are 4, 8, 12, 16, 20, 24, 28, 32, and 36. Only 4, 12, and 36 appear on both lists, so the answer is 3, which is option (B) 3.

**Solution 2:**
$36 = 4^1 \cdot 3^2$. All possible factors of 36 will be here, except for ones divisible by 2 and not by 4. $(1+1) \cdot (2+1) = 6$. Subtract factors not divisible by 4, which are 1, $3^1$, and $3^2$. 6-3=3, which is (B) 3.

**New solution:**
To solve the problem of finding how many positive factors of 36 are also multiples of 4, we can use a more systematic approach by first determining the prime factorization of 36 and then identifying the factors that are multiples of 4.

1. Prime Factorization of 36: [$36 = 2^2 \times 3^2$]
2. Identify the General Form of Factors: Any factor of 36 can be written in the form ($2^a \times 3^b$), where ($0 \le a \le 2$) and ($0 \le b \le 2$).
3. Determine the Conditions for Multiples of 4: For a factor to be a multiple of 4, it must contain at least ($2^2$) in its prime factorization. Therefore, (a) must be at least 2.
4. Possible Values for (a) and (b):
   - Since (a) must be at least 2, (a) can be either 2.
   - (b) can be 0, 1, or 2.
5. List All Possible Factors:
   - For (a = 2):
      - (b = 0): ($2^2 \times 3^0 = 4$)
      - (b = 1): ($2^2 \times 3^1 = 12$)
      - (b = 2): ($2^2 \times 3^2 = 36$)
6. Count the Factors: There are 3 factors of 36 that are also multiples of 4: (4, 12,) and (36). Therefore, the number of positive factors of 36 that are also multiples of 4 is (B).

**Please output "YES" if the new solution leads to the same results as the reference solutions; otherwise, output "NO". YES or NO?**

Figure 6: Example prompt used for correctness evaluation

**Criteria for evaluating the difference between two mathematical solutions include:**
i). If the methods used to arrive at the solutions are fundamentally different, such as algebraic manipulation versus geometric reasoning, they can be considered distinct;
ii). Even if the final results are the same, if the intermediate steps or processes involved in reaching those solutions vary significantly, the solutions can be considered different;
iii). If two solutions rely on different assumptions or conditions, they are likely to be distinct;
iv). A solution might generalize to a broader class of problems, while another solution might be specific to certain conditions. In such cases, they are considered distinct;
v). If one solution is significantly simpler or more complex than the other, they can be regarded as essentially different, even if they lead to the same result.

**Given the following mathematical problem:**
What is the largest power of 2 that is a divisor of \(13^4-11^4\)?

**Reference solutions:**
**Solution 1:**
The factors of 36 are 1, 2, 3, 4, 6, 9, 12, 18, and 36. The multiples of 4 up to 36 are 4, 8, 12, 16, 20, 24, 28, 32, and 36. Only 4, 12, and 36 appear on both lists, so the answer is 3, which is option (B) 3.

**New solution:**
To solve the problem of finding how many positive factors of 36 are also multiples of 4, we can use a more systematic approach by first determining the prime factorization of 36 and then identifying the factors that are multiples of 4.

1. Prime Factorization of 36: [$36 = 2^2 \times 3^2$]
2. Identify the General Form of Factors: Any factor of 36 can be written in the form ($2^a \times 3^b$), where ($0 \le a \le 2$) and ($0 \le b \le 2$).
3. Determine the Conditions for Multiples of 4: For a factor to be a multiple of 4, it must contain at least ($2^2$) in its prime factorization. Therefore, (a) must be at least 2.
4. Possible Values for (a) and (b):
   - Since (a) must be at least 2, (a) can be either 2.
   - (b) can be 0, 1, or 2.
5. List All Possible Factors:
   - For (a = 2):
      - (b = 0): ($2^2 \times 3^0 = 4$)
      - (b = 1): ($2^2 \times 3^1 = 12$)
      - (b = 2): ($2^2 \times 3^2 = 36$)
6. Count the Factors: There are 3 factors of 36 that are also multiples of 4: (4, 12,) and (36). Therefore, the number of positive factors of 36 that are also multiples of 4 is (B).

**Please output "YES" if the new solution is a novel solutions; otherwise, output "NO". YES or NO?**

Figure 7: Example prompt used for novelty (creativity) evaluation

**Addressing potential limitations**   We acknowledge that LLM-based evaluation may involve intrinsic biases. However, we mitigated this concern by grounding evaluations on human-written reference solutions and by explicitly requiring reasoning for each judgment, thereby reducing the risk of arbitrary or biased labeling.

### B.1.2   Criteria for Creativity Judgment

To determine creativity, we combined the results of the two LLM evaluators using a union-based criterion: if either evaluator judged a solution as creative, we labeled it as a *Creative Solution*. Since creativity inherently involves a certain degree of subjectivity, requiring agreement from both evaluators could risk overlooking genuinely creative responses. By applying the union-based criterion, we aimed to mitigate the risk of underestimating creativity.

To verify the reliability of this approach, we measured Cohen's kappa score between the two evaluators. The score was 0.741, which corresponds to substantial agreement, indicating that the two evaluators produced highly consistent judgments.

| Case 1: LLM Evaluator Output — Creative | Case 2: LLM Evaluator Output — Not Creative |
|---|---|
| The new solution uses a completely different approach (Lifting The Exponent Lemma) compared to the reference solutions (difference of squares factorization). This satisfies criterion 1 for novelty. While the final result is the same, the intermediate steps and the mathematical tools used are entirely distinct, satisfying criterion 2. The LTE lemma also relies on different underlying principles and conditions than simple algebraic manipulation, satisfying criterion 3. Therefore, the new solution is considered novel. | The new solution is not novel. It uses the same logical deductions as the reference solution. Both solutions arrive at the same answer through the same constraints. The core logic—Carl not being next to Bret, and Abby not being between Bret and Carl—is identical. The new solution merely restates these conditions and elaborates on the placement possibilities without introducing any new mathematical techniques, different assumptions, or a more generalized approach. Therefore, it does not meet any of the criteria for novelty. |

Figure 8: Example outputs from the LLM evaluators for determining creativity. The evaluators not only provide a binary judgment (Creative or Not Creative) but also justify their decision with a textual rationale based on the predefined criteria.

## B.2 LLM Generator Details

For all LLM Generators, the maximum input token length was set to 2000, and the maximum output token length was limited to 1023. Top-$p$ was fixed at 1.0, and Top-$k$ was fixed at 50 across all models. Temperature values were adjusted for each model to encourage the generation of Creative Solutions, and the final settings used for dataset construction are as follows:

- **DeepSeek-Math-7B** (`deepseek-ai/deepseek-math-7b-rl`): 0.7
- **Mathstral-7B** (`mistralai/Mathstral-7b-v0.1`): 0.25
- **OpenMath2-LLaMA3.1-8B** (`nvidia/OpenMath2-Llama3.1-8B`): 1.0
- **OREAL-7B** (`internlm/OREAL-7B`): 0.7
- **Qwen2.5-Math-7B** (`Qwen/Qwen2.5-Math-7B-Instruct`): 0.7

All generations were performed in parallel on eight NVIDIA RTX A5000 GPUs (24GB VRAM).

## B.3 Evaluation Strategy Details

We evaluate our methods using a variety of Evaluation strategies, including thresholding, distance-based prototype matching, and trainable models such as MLP, TabM, and Decision-tree based XGBOOST. The implementation and hyperparameter settings for each method are summarized below.

**Threshold (only for Baselines)**  We divide the value range of each baseline measure into 200 intervals and evaluate performance at each threshold. The threshold that achieves the best macro-f1 score on the reference set is selected for final evaluation.

**Prototype (only for CLAWS)**  We used an Encoder consisting of two Linear Layers for the prototype-based evaluation method. The input dimension is reduced to 16 dimensions through the first Linear Layer and reduced to 8 dimensions through the second Layer. After that, it is expanded to 16 dimensions again and reduced to the dimension corresponding to the final number of classes, and used as the output. The output of each data was averaged by class and used as the class center value. Afterwards, the Euclidean distance between each data sample and the class center was calculated to predict the closest class. We generated prototypes of the reference set using 20 different random seeds and presented the one that achieved the best macro-f1 score.

**XGBOOST**  For classification, we adopt the XGBoost algorithm [33], a gradient boosting framework based on ensembles of decision trees. In the hallucination detection setting, the model optimizes the logistic loss function, which corresponds to the binary cross-entropy objective. For creativity

Table 5: Comparison of feature generation using attention weights from different layers. CLAWS, leverages the last-layer attention weights, and this table shows how performance changes when attention weights are taken from other layers. Results are based on applying the Prototype strategy to responses generated by DeepSeek-Math-7B-RL and Qwen-2.5-Math-7B on the test dataset. Metrics are weighted F1 ($F1_w$), macro F1 ($F1_m$), macro average precision ($AP_m$), and AUROC.

| | **DeepSeek** | | | | | **Qwen** | | | |
|---|---|---|---|---|---|---|---|---|---|
| Layer | $F1_w$ | $F1_m$ | $AP_m$ | AUROC | Layer | $F1_w$ | $F1_m$ | $AP_m$ | AUROC |
| 5 | 57.00 | 45.02 | 40.15 | 60.58 | 5 | 48.37 | 42.05 | 37.12 | 57.82 |
| 10 | 58.03 | 45.33 | 40.42 | 61.40 | 10 | 48.12 | 42.48 | 38.01 | 58.65 |
| 15 | 56.94 | 44.76 | 39.81 | 60.22 | 15 | 50.64 | 43.31 | 39.45 | 59.10 |
| 20 | 57.89 | 45.61 | 41.08 | 61.93 | 20 | 49.85 | 42.71 | 38.74 | 58.98 |
| 25 | 57.25 | 44.98 | 40.41 | 61.37 | 25 | 50.01 | 43.25 | 39.21 | 59.26 |
| Last(30) | 58.66 | 46.01 | 41.17 | 62.09 | Last(28) | 50.35 | 43.37 | 39.88 | 59.32 |

detection, it minimizes the softmax cross-entropy loss, producing a discrete class label corresponding to the highest posterior probability.

**MLP**   We use a three-layer feed-forward neural network. The model is trained for 10 epochs using cross-entropy loss with class weights to account for class imbalance. Optimization is performed using Adam with a learning rate of 0.001. Depending on the number of classes and the input feature dimension, the model contains up to 133 learnable parameters. We experimented with 20 different random seeds and presented the one that gave the best macro-f1 score.

**TabM**   We use TabM [34], an ensemble of 32 independently parameterized MLPs. Each MLP has 512 parameters. model in the ensemble is trained for 20 epochs using cross-entropy loss. Optimization is performed using AdamW with a learning rate of 2e-3 and a weight decay of 3e-4.

## C   Experimental Results

### C.1   Layer-Wise Analysis

We present the layer-wise performance evaluation in Table 5. The results show that there are no significant differences in performance across layers, and this trend is consistent across all models and metrics. These findings indicate that CLAWS does not rely on any specific layer and operates robustly under various configurations.

### C.2   Visualizations for Reference Set

Figure 9 presents class-wise average scores across different methods on the reference set for four models. Results for Qwen-2.5-Math-7B are shown separately in Figure 4.

### C.3   Full Table of Creativity Detection

The results of 3-class detection for each model using different evaluation strategies (see Section 4.2) are presented in Tables 6–10. The results for the Threshold and Prototype-based methods are shown separately in Table 2.

### C.4   Full Table of Hallucination Detection

The results of 2-class detection for each model using different evaluation strategies are presented in Tables 11–15. The results for the Threshold and Prototype-based methods are shown separately in Table 3.

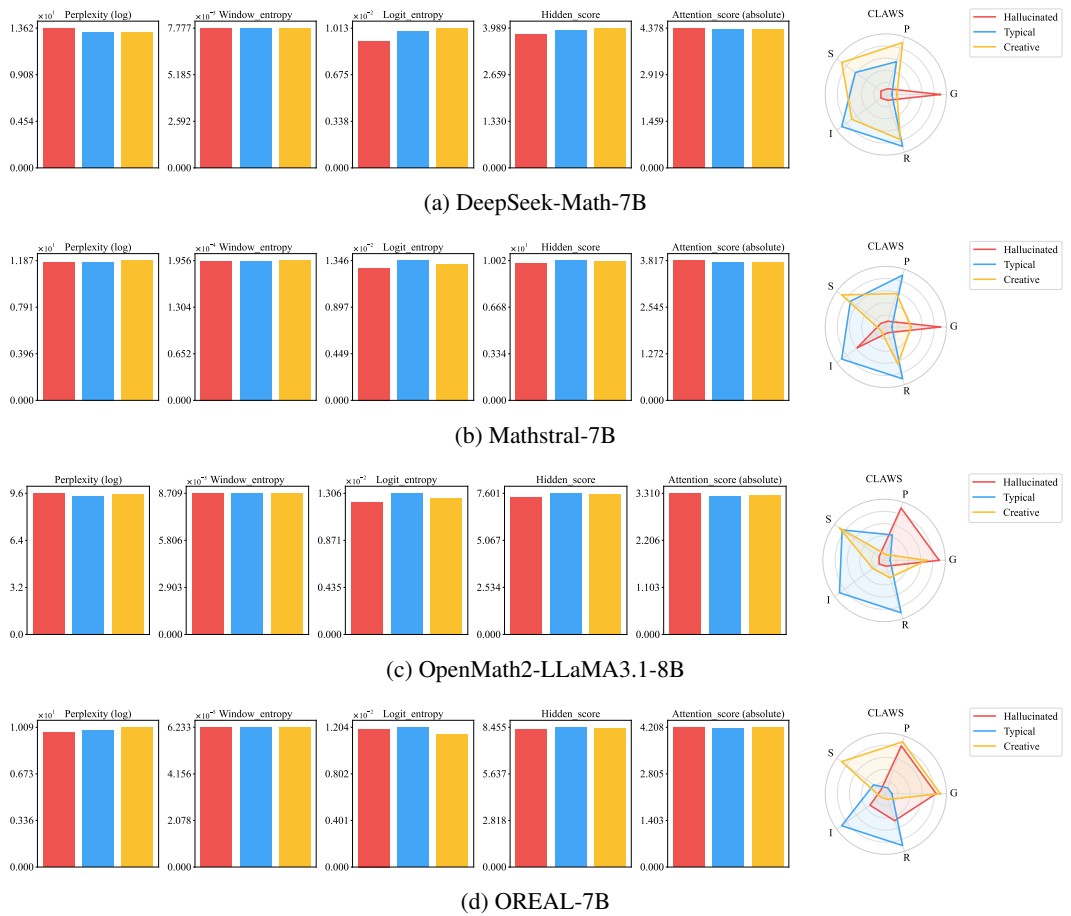

Figure 9: Visualization of class-wise average scores across all evaluation methods for four models on the reference set. For CLAWS, the scores are normalized and clipped to the range [0.1, 0.9] to enhance visual clarity.

## C.5 Full Table of Creativity Detection for Balanced Reference Set

The results of 3-class balanced detection using the Threshold and Prototype-based methods are shown in Table 16, followed by model-specific results using other evaluation strategies in Tables 17–21.

Table 6: Evaluation results for creativity detection using DeepSeek-Math-7B. Bold values indicate the best performance, underlined values indicate the second-best. Light gray-shaded cells correspond to results where the model performed detection over only two out of the three target classes, while dark gray-shaded cells indicate cases where the model predicted only one out of the three target classes.

| Dataset | | TEST | | | | AMC | | | | AIME | | | | A(J)HSME | | | |
|---|---|---|---|---|---|---|---|---|---|---|---|---|---|---|---|---|---|
| Strategy | Method | $F1_w$ | $F1_m$ | $AP_m$ | AUROC | $F1_w$ | $F1_m$ | $AP_m$ | AUROC | $F1_w$ | $F1_m$ | $AP_m$ | AUROC | $F1_w$ | $F1_m$ | $AP_m$ | AUROC |
| XGBOOST | PPL | 50.32 | 36.08 | 37.52 | 56.82 | 40.34 | 32.65 | 36.65 | 53.92 | 55.57 | 34.55 | 36.68 | 55.38 | 38.30 | 33.32 | **37.23** | 55.15 |
| | WE | **59.39** | **42.20** | **43.01** | **63.81** | 41.53 | 33.75 | 38.03 | **57.36** | 56.51 | 31.67 | 35.92 | 55.01 | 38.52 | 33.52 | 37.16 | **56.25** |
| | LE | 45.62 | 34.29 | 34.82 | 52.23 | 37.96 | 31.41 | 34.63 | 51.27 | 50.71 | 33.27 | 35.18 | 52.83 | 34.09 | 30.10 | 34.80 | 52.29 |
| | HS | 49.14 | 37.55 | 38.40 | 56.61 | 39.37 | 32.70 | 34.52 | 51.19 | 47.59 | 31.38 | 32.97 | 49.50 | 35.47 | 31.85 | 34.74 | 51.74 |
| | AS | 53.58 | 38.76 | 42.19 | 63.53 | 38.09 | 30.36 | 34.31 | 51.09 | 51.56 | 31.66 | 33.97 | 49.75 | 33.85 | 28.86 | 33.85 | 50.40 |
| | CLAWS | 57.85 | **45.59** | **47.98** | **68.39** | **44.37** | **37.67** | **38.57** | 56.99 | 51.12 | 34.53 | **38.65** | **57.31** | **39.04** | **34.49** | **37.23** | 54.88 |
| MLP | PPL | 41.04 | 31.64 | 40.85 | 59.94 | 42.58 | 33.79 | 39.38 | 56.92 | 53.72 | 35.93 | 40.08 | 61.61 | 36.36 | 30.81 | 39.57 | 57.42 |
| | WE | 58.29 | 42.05 | 42.32 | 62.73 | **44.30** | **35.70** | 37.58 | 56.56 | 56.51 | 31.67 | 35.90 | 54.68 | 38.62 | 33.61 | 36.95 | 55.99 |
| | LE | 39.99 | 32.84 | 35.47 | 52.45 | 41.30 | 35.43 | 35.48 | 52.18 | 46.20 | 30.22 | 35.41 | 53.91 | 38.85 | 35.80 | 36.67 | 53.31 |
| | HS | 48.79 | 41.59 | 42.00 | 62.98 | 35.61 | 27.90 | 34.76 | 52.59 | 35.63 | 28.31 | 32.55 | 48.47 | 34.19 | 32.00 | 34.58 | 50.59 |
| | AS | 51.05 | 37.63 | 43.02 | 64.31 | 36.61 | 28.87 | 33.88 | 50.68 | 46.90 | 30.25 | 34.07 | 49.98 | 32.78 | 27.70 | 33.63 | 49.79 |
| | CLAWS | **61.43** | **49.65** | 50.08 | **71.41** | 43.78 | 34.84 | **41.01** | **60.11** | **57.39** | **39.88** | **42.26** | **62.56** | **44.11** | **39.65** | **42.64** | **60.12** |
| TabM | PPL | 51.79 | 37.78 | 41.03 | 60.03 | 43.76 | 35.20 | 39.29 | 56.83 | 56.84 | 35.80 | 41.01 | 61.73 | 40.89 | 35.30 | 39.67 | 57.42 |
| | WE | 59.39 | 42.20 | 42.93 | 63.27 | 41.53 | 33.75 | 38.00 | 57.13 | 56.51 | 31.67 | 36.16 | 54.97 | 38.62 | 33.61 | 37.17 | 56.13 |
| | LE | 42.80 | 31.66 | 35.39 | 53.49 | 40.44 | 32.32 | 35.45 | 51.89 | 48.06 | 30.82 | 36.39 | 54.66 | 37.27 | 32.04 | 36.73 | 53.64 |
| | HS | 51.25 | 36.33 | 42.64 | 63.04 | 38.75 | 31.17 | 34.86 | 52.60 | 48.77 | 29.56 | 32.40 | 48.14 | 34.58 | 29.99 | 35.34 | 51.85 |
| | AS | 51.05 | 37.63 | 42.83 | 64.13 | 36.61 | 28.87 | 33.97 | 50.77 | 46.90 | 30.25 | 33.76 | 49.82 | 32.78 | 27.70 | 33.68 | 49.85 |
| | CLAWS | **60.78** | **45.03** | **51.37** | **72.82** | **46.46** | **37.56** | **41.06** | **59.26** | **57.05** | **37.77** | **41.44** | 60.73 | **40.81** | 35.10 | **41.25** | **59.12** |

Table 7: Evaluation results for creativity detection using Mathstral-7B

| Dataset | | TEST | | | | AMC | | | | AIME | | | | A(J)HSME | | | |
|---|---|---|---|---|---|---|---|---|---|---|---|---|---|---|---|---|---|
| Strategy | Method | $F1_w$ | $F1_m$ | $AP_m$ | AUROC | $F1_w$ | $F1_m$ | $AP_m$ | AUROC | $F1_w$ | $F1_m$ | $AP_m$ | AUROC | $F1_w$ | $F1_m$ | $AP_m$ | AUROC |
| XGBOOST | PPL | 55.02 | 31.81 | 36.77 | 56.43 | 41.89 | 29.03 | 34.58 | 51.87 | 64.84 | 31.95 | 34.91 | 54.42 | 38.87 | 28.83 | 35.90 | 53.20 |
| | WE | 58.25 | 34.05 | **44.34** | **67.71** | 45.13 | 31.05 | 41.15 | **61.71** | 65.90 | 29.74 | 35.19 | 54.33 | 41.84 | 31.38 | 39.03 | 58.88 |
| | LE | 52.67 | 29.93 | 33.46 | 49.70 | 43.28 | 30.38 | 36.59 | 54.33 | 63.76 | 31.62 | 32.94 | 49.05 | 37.13 | 27.49 | 34.06 | 51.18 |
| | HS | 59.40 | 38.04 | 41.71 | 62.34 | 45.00 | 34.05 | 37.12 | 54.68 | 64.85 | 30.67 | 32.82 | 49.71 | 38.97 | 29.97 | 35.07 | 52.29 |
| | AS | 57.64 | 34.26 | 41.54 | 60.55 | 41.61 | 28.15 | 39.25 | 57.78 | 65.34 | 28.87 | 33.94 | 49.11 | 38.13 | 28.34 | 36.23 | 53.93 |
| | CLAWS | **62.05** | **42.09** | 44.25 | 64.43 | **50.92** | **38.85** | **40.31** | 57.69 | **69.30** | **40.42** | **41.21** | **59.53** | **47.04** | **38.75** | **40.66** | **59.06** |
| MLP | PPL | 53.42 | 32.71 | 39.61 | 59.24 | 36.93 | 28.62 | 36.25 | 52.83 | 54.76 | 33.05 | 34.34 | 52.57 | 34.71 | 26.01 | 35.11 | 51.70 |
| | WE | 49.45 | 28.16 | 37.02 | 55.53 | 37.14 | 23.20 | 35.41 | 52.17 | 65.47 | 28.70 | 34.48 | 52.90 | 33.06 | 22.45 | 33.79 | 49.80 |
| | LE | 49.54 | 32.89 | 37.04 | 56.02 | 46.20 | 34.91 | 36.55 | 54.41 | 57.77 | 31.25 | 32.55 | 48.89 | 38.70 | 31.36 | 34.14 | 51.37 |
| | HS | 60.32 | 40.84 | 45.48 | 68.58 | 38.35 | 30.42 | 39.90 | 58.59 | 63.39 | 32.59 | 33.02 | 50.08 | 38.05 | 30.74 | 35.76 | 53.95 |
| | AS | 62.82 | 42.48 | 44.00 | 63.55 | 49.70 | 37.03 | 38.63 | 58.10 | 61.26 | 33.63 | 33.59 | 50.32 | 41.86 | 33.38 | 37.48 | 55.85 |
| | CLAWS | **64.65** | **45.34** | **47.74** | **70.25** | **52.01** | **41.33** | **41.89** | **61.03** | **69.54** | **39.70** | **41.51** | **61.86** | **45.81** | **40.58** | **42.67** | **60.58** |
| TabM | PPL | 52.72 | 26.55 | 38.26 | 59.43 | 37.14 | 23.20 | 38.09 | 56.46 | 65.69 | 28.80 | 35.89 | 54.48 | 39.68 | 28.93 | 32.67 | 48.80 |
| | WE | 52.72 | 26.55 | 37.06 | 55.54 | 37.72 | 23.78 | 35.59 | 52.25 | 66.14 | 29.66 | 34.96 | 53.40 | 45.45 | 34.94 | 35.90 | 53.75 |
| | LE | 52.39 | 26.74 | 35.96 | 54.93 | 37.81 | 23.92 | 36.26 | 53.96 | 65.45 | 29.51 | 32.01 | 47.47 | 41.61 | 31.58 | 34.33 | 50.45 |
| | HS | **64.38** | **42.69** | 45.11 | **68.38** | 48.44 | 35.81 | 39.97 | 58.39 | 65.09 | 29.99 | 32.99 | 50.08 | 48.20 | 36.14 | 39.90 | 57.52 |
| | AS | 58.04 | 33.81 | 43.98 | 66.28 | 42.35 | 28.54 | 39.84 | 58.55 | 66.10 | 30.22 | 34.19 | 51.11 | 45.81 | 34.68 | 36.07 | 52.79 |
| | CLAWS | 62.96 | 43.67 | 45.16 | 65.28 | 50.44 | 38.13 | 40.67 | 58.14 | 68.05 | 37.58 | 40.03 | 58.42 | 54.29 | 41.03 | 46.14 | 65.47 |

Table 8: Evaluation results for creativity detection using OpenMath2-LLaMA3.1-8B

| Dataset | | TEST | | | | AMC | | | | AIME | | | | A(J)HSME | | | |
|---|---|---|---|---|---|---|---|---|---|---|---|---|---|---|---|---|---|
| Strategy | Method | F1$_w$ | F1$_m$ | AP$_m$ | AUROC | F1$_w$ | F1$_m$ | AP$_m$ | AUROC | F1$_w$ | F1$_m$ | AP$_m$ | AUROC | F1$_w$ | F1$_m$ | AP$_m$ | AUROC |
| XGBOOST | PPL | 47.78 | 33.14 | 34.54 | 52.60 | 42.29 | 31.00 | 33.16 | 49.65 | 46.38 | 32.22 | 32.55 | 50.81 | 40.45 | 31.02 | 34.02 | 51.25 |
| | WE | 58.84 | 40.14 | 43.09 | 63.58 | 49.28 | 36.50 | 40.17 | 58.33 | 43.16 | 28.33 | 35.66 | 52.80 | 47.00 | 36.16 | 38.35 | 57.20 |
| | LE | 47.12 | 33.47 | 34.59 | 50.79 | 44.35 | 33.34 | 34.88 | 52.15 | 44.58 | 30.92 | 33.01 | 48.48 | 39.81 | 30.93 | 33.49 | 49.91 |
| | HS | 54.90 | 38.46 | 41.22 | 60.99 | 48.40 | 35.39 | 37.51 | 54.81 | 48.65 | 34.10 | 34.91 | 51.78 | 45.77 | 35.26 | 37.85 | 55.67 |
| | AS | 57.43 | 40.10 | 42.07 | 61.57 | 49.95 | 36.40 | 37.26 | 54.48 | 48.73 | 33.54 | 35.32 | 50.98 | 44.56 | 33.59 | 35.40 | 52.01 |
| | CLAWS | 63.20 | 46.23 | 48.16 | 67.22 | 54.56 | 40.24 | 42.68 | 61.61 | 56.87 | 40.82 | 42.51 | 59.67 | 50.40 | 38.13 | 42.63 | 61.47 |
| MLP | PPL | 32.70 | 24.95 | 31.40 | 47.66 | 40.22 | 28.52 | 32.40 | 50.26 | 37.94 | 27.23 | 30.08 | 44.11 | 39.52 | 28.74 | 33.17 | 51.08 |
| | WE | 54.52 | 38.02 | 37.93 | 57.74 | 45.19 | 32.85 | 35.74 | 52.96 | 36.87 | 23.13 | 33.27 | 49.26 | 43.96 | 33.30 | 34.99 | 51.99 |
| | LE | 37.41 | 27.93 | 33.52 | 48.99 | 42.73 | 30.66 | 32.41 | 47.40 | 41.31 | 29.13 | 30.58 | 44.97 | 41.39 | 30.77 | 33.39 | 49.88 |
| | HS | 57.11 | 40.73 | 46.31 | 66.72 | 50.02 | 36.02 | 40.98 | 58.74 | 49.78 | 35.33 | 36.12 | 52.65 | 48.00 | 35.80 | 39.40 | 56.58 |
| | AS | 57.30 | 40.33 | 42.38 | 63.12 | 50.74 | 36.76 | 37.99 | 54.72 | 49.81 | 34.53 | 35.05 | 51.70 | 46.34 | 34.75 | 35.91 | 52.32 |
| | CLAWS | 65.32 | 47.69 | 52.80 | 73.42 | 58.31 | 43.82 | 48.59 | 67.67 | 58.11 | 42.09 | 44.17 | 61.44 | 52.72 | 42.31 | 45.52 | 64.63 |
| TabM | PPL | 35.59 | 26.46 | 31.59 | 47.64 | 40.66 | 28.92 | 32.32 | 49.46 | 39.10 | 27.92 | 29.78 | 43.42 | 32.90 | 22.11 | 35.43 | 52.59 |
| | WE | 58.46 | 40.05 | 40.11 | 60.38 | 48.15 | 35.64 | 36.65 | 54.54 | 45.85 | 30.55 | 35.01 | 51.67 | 33.20 | 22.42 | 34.08 | 50.31 |
| | LE | 47.65 | 33.30 | 35.52 | 52.44 | 45.31 | 33.11 | 34.86 | 52.12 | 44.65 | 31.01 | 31.60 | 45.48 | 33.40 | 22.78 | 35.11 | 52.27 |
| | HS | 58.51 | 41.39 | 46.16 | 66.42 | 52.12 | 37.80 | 41.05 | 58.90 | 50.78 | 35.73 | 36.14 | 52.56 | 41.22 | 31.66 | 36.10 | 54.01 |
| | AS | 57.09 | 39.55 | 42.58 | 63.30 | 51.33 | 37.50 | 38.17 | 55.22 | 47.05 | 32.05 | 35.16 | 51.83 | 39.04 | 28.53 | 37.16 | 55.69 |
| | CLAWS | 66.45 | 46.83 | 52.95 | 73.17 | 58.70 | 42.78 | 47.83 | 66.93 | 58.92 | 40.95 | 44.10 | 60.67 | 54.20 | 41.15 | 46.42 | 64.95 |

Table 9: Evaluation results for creativity detection using OREAL-7B

| Dataset | | TEST | | | | AMC | | | | AIME | | | | A(J)HSME | | | |
|---|---|---|---|---|---|---|---|---|---|---|---|---|---|---|---|---|---|
| Strategy | Method | F1$_w$ | F1$_m$ | AP$_m$ | AUROC | F1$_w$ | F1$_m$ | AP$_m$ | AUROC | F1$_w$ | F1$_m$ | AP$_m$ | AUROC | F1$_w$ | F1$_m$ | AP$_m$ | AUROC |
| XGBOOST | PPL | 56.08 | 31.31 | 35.98 | 54.06 | 51.46 | 30.83 | 36.59 | 56.38 | 72.38 | 33.10 | 33.81 | 50.45 | 44.41 | 29.62 | 35.92 | 53.99 |
| | WE | 59.33 | 33.74 | 36.28 | 54.38 | 55.08 | 33.66 | 36.80 | 55.13 | 73.10 | 30.26 | 33.47 | 51.08 | 50.95 | 34.42 | 37.19 | 54.80 |
| | LE | 54.29 | 28.42 | 33.05 | 48.22 | 50.04 | 28.78 | 33.53 | 50.16 | 72.80 | 32.68 | 33.59 | 50.18 | 43.16 | 27.57 | 33.80 | 51.14 |
| | HS | 61.54 | 38.44 | 39.26 | 58.61 | 57.30 | 38.33 | 38.29 | 56.62 | 70.42 | 33.57 | 33.90 | 53.02 | 52.60 | 36.87 | 37.58 | 55.27 |
| | AS | 53.83 | 26.76 | 36.00 | 53.50 | 47.70 | 25.56 | 35.93 | 53.68 | 73.07 | 29.92 | 34.63 | 55.29 | 39.22 | 23.70 | 34.03 | 50.52 |
| | CLAWS | 62.41 | 40.83 | 42.20 | 62.33 | 54.03 | 37.02 | 38.32 | 57.29 | 70.94 | 33.49 | 33.87 | 49.05 | 49.02 | 35.11 | 36.37 | 54.38 |
| MLP | PPL | 59.88 | 35.79 | 37.54 | 56.31 | 58.17 | 41.28 | 41.52 | 62.85 | 68.61 | 35.39 | 35.91 | 57.86 | 48.78 | 36.16 | 41.35 | 59.53 |
| | WE | 53.83 | 26.76 | 33.11 | 49.04 | 47.70 | 25.56 | 32.56 | 49.16 | 73.07 | 29.92 | 33.51 | 51.24 | 39.22 | 23.70 | 34.50 | 50.36 |
| | LE | 53.76 | 26.72 | 33.25 | 49.11 | 47.70 | 25.56 | 33.25 | 51.68 | 73.07 | 29.92 | 34.31 | 52.45 | 46.63 | 31.35 | 35.59 | 53.16 |
| | HS | 57.25 | 37.18 | 39.90 | 58.83 | 48.60 | 34.10 | 39.34 | 56.63 | 62.86 | 31.67 | 33.29 | 50.56 | 39.22 | 23.70 | 33.37 | 50.04 |
| | AS | 62.98 | 38.54 | 39.78 | 55.90 | 54.91 | 36.53 | 37.32 | 55.57 | 68.89 | 33.82 | 34.24 | 52.77 | 49.86 | 35.87 | 36.03 | 53.61 |
| | CLAWS | 60.12 | 41.87 | 42.39 | 61.48 | 51.30 | 37.14 | 40.19 | 59.47 | 66.67 | 34.20 | 33.83 | 50.25 | 48.76 | 37.00 | 41.53 | 59.78 |
| TabM | PPL | 53.83 | 26.76 | 37.84 | 57.08 | 47.70 | 25.56 | 38.36 | 58.89 | 73.07 | 29.92 | 35.26 | 56.47 | 39.22 | 23.70 | 38.30 | 57.69 |
| | WE | 59.16 | 34.17 | 35.25 | 52.75 | 55.08 | 33.66 | 36.22 | 54.09 | 73.10 | 30.26 | 33.35 | 50.55 | 50.95 | 34.42 | 36.90 | 54.54 |
| | LE | 53.83 | 26.76 | 31.97 | 46.90 | 47.70 | 25.56 | 32.06 | 49.43 | 73.07 | 29.92 | 32.20 | 47.67 | 39.22 | 23.70 | 34.54 | 52.61 |
| | HS | 62.28 | 38.75 | 40.26 | 58.73 | 57.60 | 38.44 | 39.42 | 57.29 | 71.32 | 33.38 | 33.59 | 51.54 | 52.55 | 36.74 | 38.95 | 56.34 |
| | AS | 53.83 | 26.76 | 39.79 | 58.17 | 47.70 | 25.56 | 37.33 | 55.62 | 73.07 | 29.92 | 33.94 | 49.19 | 39.22 | 23.70 | 39.28 | 56.80 |
| | CLAWS | 62.86 | 39.09 | 42.41 | 63.48 | 55.53 | 37.45 | 38.75 | 58.23 | 71.21 | 33.92 | 35.08 | 51.07 | 51.27 | 36.49 | 38.07 | 57.23 |

Table 10: Evaluation results for creativity detection using Qwen-2.5-Math-7B

| Dataset | | TEST | | | | AMC | | | | AIME | | | | A(J)HSME | | | |
|---|---|---|---|---|---|---|---|---|---|---|---|---|---|---|---|---|---|
| Strategy | Method | $F1_w$ | $F1_m$ | $AP_m$ | AUROC | $F1_w$ | $F1_m$ | $AP_m$ | AUROC | $F1_w$ | $F1_m$ | $AP_m$ | AUROC | $F1_w$ | $F1_m$ | $AP_m$ | AUROC |
| XGBOOST | PPL | 48.70 | 41.51 | 41.49 | 59.98 | 46.82 | 37.65 | 36.90 | 54.52 | 44.53 | 36.78 | 37.82 | 55.39 | 44.21 | 34.63 | 35.26 | 53.28 |
| | WE | 49.96 | 39.15 | 43.47 | 63.00 | 49.93 | 36.57 | 38.63 | 57.69 | 41.13 | 32.17 | 36.10 | 52.48 | 45.39 | 32.74 | 36.82 | 55.51 |
| | LE | 38.22 | 32.42 | 33.83 | 50.94 | 40.59 | 31.06 | 34.20 | 51.68 | 35.19 | 28.98 | 33.38 | 49.97 | 37.81 | 28.36 | 34.17 | 50.89 |
| | HS | 46.98 | 39.95 | 41.80 | 57.98 | 48.68 | 37.94 | 39.70 | 58.48 | 41.31 | 33.40 | **50.87** | 34.65 | 44.80 | 33.46 | 36.41 | 54.44 |
| | AS | 42.60 | 35.24 | 37.91 | 55.60 | 42.92 | 32.76 | 35.51 | 53.54 | 38.85 | 30.92 | 33.74 | 49.56 | 40.00 | 29.96 | 34.70 | 52.49 |
| | CLAWS | **52.35** | **43.33** | **47.72** | **65.98** | **50.30** | **38.98** | **40.66** | **61.11** | **45.18** | **38.95** | 39.75 | **55.86** | **47.54** | **36.02** | **39.41** | **59.45** |
| MLP | PPL | 54.27 | **46.34** | 47.90 | 67.59 | 50.01 | 38.01 | **43.58** | 61.96 | **47.58** | 36.13 | **41.91** | **62.02** | 44.92 | **37.00** | 38.23 | **57.72** |
| | WE | 45.67 | 39.58 | 38.31 | 54.97 | 43.32 | 32.95 | 36.49 | 53.87 | 40.26 | 30.78 | 34.22 | 51.26 | 42.11 | 33.31 | 34.35 | 50.53 |
| | LE | 29.48 | 23.10 | 28.27 | 41.43 | 36.53 | 26.33 | 33.96 | 49.47 | 31.62 | 25.13 | 31.19 | 47.00 | 32.30 | 23.38 | 35.14 | 50.73 |
| | HS | **54.58** | 42.77 | 46.98 | 65.86 | 50.29 | 36.64 | 41.90 | 60.61 | 43.11 | 32.96 | 36.01 | 52.38 | 46.24 | 36.21 | 37.80 | 56.01 |
| | AS | 43.80 | 38.05 | 40.62 | 59.13 | 43.04 | 32.73 | 35.52 | 52.88 | 39.07 | 30.08 | 32.98 | 47.71 | 39.86 | 30.64 | 34.43 | 51.33 |
| | CLAWS | 53.78 | 45.32 | **50.44** | **67.98** | **52.51** | **39.65** | 43.06 | **63.20** | 42.76 | **36.59** | 39.32 | 56.49 | **53.75** | **41.05** | **42.29** | **62.61** |
| TabM | PPL | **54.53** | 42.74 | **47.13** | 65.44 | **50.83** | 38.45 | **43.07** | 60.61 | **48.30** | 37.01 | **43.37** | **60.80** | 45.58 | 34.28 | 38.54 | 57.10 |
| | WE | 48.15 | 37.73 | 40.35 | 58.70 | 46.62 | 33.89 | 36.59 | 55.06 | 38.66 | 30.12 | 34.82 | 51.49 | 45.58 | 31.17 | 34.81 | 52.06 |
| | LE | 33.24 | 26.04 | 32.55 | 50.24 | 41.43 | 28.20 | 34.61 | 52.39 | 31.50 | 25.36 | 33.98 | 50.30 | 38.81 | 25.93 | 35.89 | 52.41 |
| | HS | 51.47 | 41.39 | 45.50 | 63.64 | 50.08 | 36.79 | 41.58 | **61.00** | 43.75 | 34.48 | 36.73 | 53.74 | 46.01 | 33.00 | 37.82 | 55.41 |
| | AS | 45.76 | 35.86 | 38.65 | 56.76 | 43.53 | 31.74 | 36.03 | 54.26 | 37.84 | 29.68 | 33.06 | 49.16 | 41.37 | 28.96 | 35.60 | 54.28 |
| | CLAWS | 51.93 | **42.92** | 45.82 | 63.83 | 48.69 | **38.90** | 39.23 | 59.61 | 45.14 | **38.28** | 38.15 | 55.26 | 47.31 | 35.97 | 39.23 | 59.28 |

Table 11: Evaluation results for hallucination detection using DeepSeek-Math-7B. Bold and underlined values indicate the best and second-best performance, respectively. Light gray-shaded cells indicate cases where the model predicted only a single class in a 2-class detection setting.

| Dataset | | TEST | | | | AMC | | | | AIME | | | | A(J)HSME | | | |
|---|---|---|---|---|---|---|---|---|---|---|---|---|---|---|---|---|---|
| Strategy | Method | $F1_w$ | $F1_m$ | $AP_m$ | AUROC | $F1_w$ | $F1_m$ | $AP_m$ | AUROC | $F1_w$ | $F1_m$ | $AP_m$ | AUROC | $F1_w$ | $F1_m$ | $AP_m$ | AUROC |
| XGBOOST | PPL | 56.31 | 56.08 | 52.18 | 59.45 | 54.76 | 53.53 | 66.15 | 55.92 | 57.65 | **53.32** | 40.16 | 56.75 | 58.61 | 56.18 | 70.04 | 58.75 |
| | WE | 65.53 | 63.87 | 60.55 | 68.13 | 50.79 | 52.21 | **68.78** | 60.84 | **59.51** | 49.52 | 38.21 | 55.66 | 53.27 | 53.41 | 70.15 | 59.51 |
| | LE | 50.45 | 50.49 | 44.87 | 52.12 | 52.12 | 50.32 | 62.19 | 51.76 | 53.25 | 50.66 | 36.64 | 53.36 | 54.86 | 52.15 | 65.98 | 53.49 |
| | HS | 57.91 | 57.84 | 49.82 | 58.86 | 53.23 | 49.99 | 61.94 | 50.51 | 46.49 | 44.95 | 31.93 | 46.78 | 56.00 | 51.74 | 64.72 | 50.91 |
| | AS | 61.26 | 61.37 | 57.39 | 67.27 | 53.47 | 49.26 | 61.88 | 50.82 | 52.64 | 48.97 | 34.84 | 50.44 | 55.74 | 49.52 | 64.52 | 49.93 |
| | CLAWS | **66.15** | **66.02** | **66.06** | **73.21** | **58.95** | **56.00** | 68.65 | 59.39 | 55.34 | 52.80 | **43.24** | **58.22** | **61.12** | **56.83** | **71.01** | **59.99** |
| MLP | PPL | 53.26 | 54.05 | 60.78 | 63.40 | 58.89 | 55.25 | 57.58 | 60.10 | 59.93 | **57.91** | 60.78 | **63.60** | 63.44 | 58.87 | 59.02 | 61.55 |
| | WE | 65.53 | 63.87 | 62.80 | 65.76 | 50.79 | 52.21 | 56.97 | 60.84 | **60.76** | 54.86 | 53.73 | 55.66 | 57.90 | 56.12 | 55.34 | 59.09 |
| | LE | 47.54 | 48.57 | 50.43 | 50.62 | 55.57 | 53.13 | 53.23 | 53.82 | 53.84 | 51.23 | 52.66 | 53.65 | 58.38 | 54.61 | 54.77 | 55.73 |
| | HS | 61.89 | 62.14 | 63.48 | 66.54 | 53.86 | 48.59 | 52.13 | 52.39 | 42.56 | 42.32 | 48.47 | 46.41 | 55.68 | 49.34 | 50.96 | 51.83 |
| | AS | 61.26 | 61.37 | 62.99 | 67.08 | 53.47 | 49.26 | 50.75 | 50.82 | 52.64 | 48.97 | 50.77 | 50.12 | 55.74 | 49.52 | 50.03 | 49.90 |
| | CLAWS | **70.69** | **70.51** | **76.44** | **78.35** | **63.48** | **61.29** | **62.59** | **65.67** | 60.50 | 57.34 | **61.15** | 62.70 | **65.12** | **61.11** | **61.64** | **64.64** |
| TabM | PPL | 59.01 | 58.98 | 54.53 | 63.40 | 58.42 | 57.20 | 69.78 | 60.62 | **62.95** | **58.42** | 45.75 | **63.60** | 62.21 | 59.47 | 71.05 | 61.55 |
| | WE | 65.53 | 63.87 | 60.57 | 68.13 | 50.79 | 52.21 | 68.78 | 60.84 | 59.51 | 49.52 | 38.17 | 55.64 | 53.46 | 53.58 | 70.20 | 59.56 |
| | LE | 47.53 | 48.49 | 41.98 | 50.62 | 55.79 | 52.92 | 63.51 | 53.82 | 52.33 | 50.51 | 36.12 | 53.71 | 58.74 | 54.71 | 66.79 | 55.29 |
| | HS | 60.90 | 61.22 | 54.83 | 65.20 | 53.59 | 48.45 | 63.08 | 52.67 | 42.46 | 42.23 | 33.84 | 46.98 | 55.16 | 48.54 | 67.30 | 53.87 |
| | AS | 62.95 | 62.43 | 57.29 | 67.33 | 53.15 | 50.65 | 61.96 | 50.82 | 56.37 | 50.39 | 35.48 | 50.12 | 54.40 | 49.92 | 64.47 | 49.94 |
| | CLAWS | **70.80** | **70.40** | **68.96** | **77.03** | **61.18** | **58.52** | **71.23** | **62.34** | 61.27 | 57.73 | **46.94** | 63.37 | **64.01** | **59.84** | **74.78** | **64.70** |

Table 12: Evaluation results for hallucination detection using Mathstral-7B

| Dataset | | TEST | | | | AMC | | | | AIME | | | | A(J)HSME | | | |
|---|---|---|---|---|---|---|---|---|---|---|---|---|---|---|---|---|---|
| Strategy | Method | F1$_w$ | F1$_m$ | AP$_m$ | AUROC | F1$_w$ | F1$_m$ | AP$_m$ | AUROC | F1$_w$ | F1$_m$ | AP$_m$ | AUROC | F1$_w$ | F1$_m$ | AP$_m$ | AUROC |
| XGBOOST | PPL | 59.16 | 51.63 | 39.99 | 57.91 | 48.13 | 46.97 | 48.19 | 52.02 | 66.72 | 53.03 | 25.93 | 53.45 | 47.97 | 48.07 | 54.48 | 53.54 |
| | WE | 61.35 | 52.24 | 52.44 | 70.96 | 48.34 | 46.73 | 59.67 | 65.49 | 66.05 | 44.75 | 26.79 | 54.60 | 47.97 | 48.12 | 59.81 | 62.04 |
| | LE | 55.98 | 48.86 | 33.54 | 50.88 | 49.70 | 48.54 | 50.70 | 54.97 | 63.63 | 47.62 | 22.38 | 47.37 | 45.96 | 46.08 | 50.91 | 51.38 |
| | HS | 64.41 | 58.97 | 49.50 | 66.57 | 53.31 | 52.71 | 52.16 | 57.07 | 65.91 | 48.86 | 23.06 | 48.36 | 50.24 | 50.32 | 53.49 | 53.79 |
| | AS | 63.29 | 54.88 | 52.28 | 66.47 | 47.93 | 46.39 | 56.53 | 61.52 | 66.37 | 46.10 | 25.84 | 52.21 | 45.81 | 45.96 | 56.78 | 56.92 |
| | CLAWS | 68.93 | 64.74 | 54.22 | 70.14 | 57.86 | 57.42 | 57.63 | 60.81 | 68.62 | 55.25 | 34.92 | 59.35 | 56.64 | 56.66 | 61.68 | 61.07 |
| MLP | PPL | 62.44 | 58.35 | 59.79 | 63.82 | 55.80 | 55.65 | 57.72 | 58.83 | 58.87 | 51.43 | 52.67 | 55.24 | 52.06 | 52.02 | 53.64 | 54.01 |
| | WE | 52.42 | 44.27 | 55.67 | 56.78 | 38.24 | 35.98 | 53.10 | 52.71 | 66.08 | 44.36 | 52.01 | 53.63 | 50.57 | 50.63 | 51.02 | 50.27 |
| | LE | 54.35 | 53.02 | 55.60 | 58.33 | 55.17 | 55.27 | 55.44 | 57.35 | 50.54 | 43.69 | 48.03 | 46.99 | 52.10 | 52.04 | 52.61 | 54.60 |
| | HS | 68.29 | 65.92 | 69.13 | 72.56 | 53.29 | 53.85 | 59.92 | 61.18 | 65.09 | 51.03 | 49.89 | 49.29 | 50.87 | 50.79 | 53.46 | 54.78 |
| | AS | 69.28 | 65.39 | 65.30 | 68.56 | 58.59 | 58.45 | 59.61 | 62.04 | 63.78 | 52.56 | 50.74 | 51.46 | 55.17 | 55.20 | 56.01 | 57.89 |
| | CLAWS | 72.60 | 70.23 | 71.58 | 76.46 | 63.97 | 63.91 | 64.43 | 67.13 | 69.36 | 55.98 | 60.50 | 65.42 | 57.48 | 57.41 | 61.13 | 63.78 |
| TabM | PPL | 52.72 | 39.83 | 41.56 | 63.13 | 37.14 | 34.80 | 52.24 | 58.69 | 65.69 | 43.19 | 28.13 | 55.72 | 32.90 | 33.16 | 51.18 | 52.35 |
| | WE | 53.00 | 40.24 | 48.29 | 67.44 | 38.28 | 36.03 | 55.86 | 62.65 | 66.08 | 44.36 | 26.90 | 54.81 | 33.56 | 33.82 | 55.86 | 56.94 |
| | LE | 53.17 | 41.09 | 37.32 | 57.25 | 38.42 | 36.18 | 51.03 | 56.34 | 65.43 | 43.89 | 21.57 | 46.00 | 34.25 | 34.49 | 52.13 | 53.35 |
| | HS | 70.78 | 65.92 | 55.96 | 69.92 | 58.19 | 57.72 | 54.09 | 59.79 | 65.07 | 44.02 | 24.20 | 51.09 | 50.56 | 50.65 | 54.18 | 54.76 |
| | AS | 65.28 | 57.77 | 54.07 | 69.09 | 52.23 | 50.99 | 57.18 | 62.07 | 66.85 | 47.91 | 25.53 | 51.90 | 49.70 | 49.82 | 56.85 | 57.84 |
| | CLAWS | 71.24 | 66.85 | 58.34 | 74.90 | 59.83 | 59.33 | 57.96 | 62.41 | 70.49 | 54.75 | 34.28 | 59.23 | 58.69 | 58.70 | 60.09 | 61.38 |

Table 13: Evaluation results for hallucination detection using OpenMath2-LLaMA3.1-8B

| Dataset | | TEST | | | | AMC | | | | AIME | | | | A(J)HSME | | | |
|---|---|---|---|---|---|---|---|---|---|---|---|---|---|---|---|---|---|
| Strategy | Method | F1$_w$ | F1$_m$ | AP$_m$ | AUROC | F1$_w$ | F1$_m$ | AP$_m$ | AUROC | F1$_w$ | F1$_m$ | AP$_m$ | AUROC | F1$_w$ | F1$_m$ | AP$_m$ | AUROC |
| XGBOOST | PPL | 50.29 | 50.23 | 45.17 | 51.55 | 49.68 | 48.63 | 57.68 | 48.70 | 48.47 | 48.42 | 46.56 | 48.88 | 50.25 | 48.29 | 62.56 | 49.13 |
| | WE | 63.59 | 62.14 | 61.51 | 68.35 | 54.17 | 55.17 | 68.09 | 63.29 | 43.86 | 42.25 | 50.17 | 54.96 | 54.85 | 55.10 | 69.61 | 60.47 |
| | LE | 51.35 | 51.07 | 46.90 | 53.15 | 51.61 | 50.97 | 59.25 | 51.71 | 47.46 | 47.15 | 45.91 | 47.80 | 48.39 | 46.90 | 60.89 | 47.56 |
| | HS | 60.00 | 59.81 | 54.59 | 64.55 | 56.74 | 55.33 | 64.64 | 57.95 | 51.68 | 51.65 | 50.50 | 52.95 | 56.39 | 53.87 | 68.48 | 57.27 |
| | AS | 63.52 | 63.06 | 57.72 | 66.34 | 58.27 | 57.02 | 63.49 | 57.37 | 51.99 | 51.46 | 49.91 | 54.12 | 56.36 | 53.87 | 64.44 | 53.77 |
| | CLAWS | 67.70 | 67.18 | 67.37 | 72.71 | 62.63 | 61.48 | 69.63 | 64.47 | 59.28 | 58.87 | 58.90 | 62.40 | 62.85 | 60.48 | 73.57 | 64.44 |
| MLP | PPL | 40.48 | 41.94 | 46.39 | 44.60 | 48.50 | 45.26 | 49.84 | 49.79 | 41.54 | 42.24 | 44.95 | 42.76 | 50.68 | 45.38 | 45.14 | 43.00 |
| | WE | 59.94 | 59.40 | 58.43 | 61.08 | 54.28 | 53.64 | 53.62 | 55.24 | 51.44 | 51.03 | 52.21 | 52.93 | 54.54 | 52.81 | 52.34 | 53.08 |
| | LE | 45.18 | 46.17 | 53.63 | 54.10 | 52.39 | 50.32 | 51.29 | 51.79 | 44.98 | 45.19 | 46.27 | 44.54 | 53.68 | 49.98 | 49.79 | 49.22 |
| | HS | 64.07 | 64.21 | 69.38 | 71.21 | 60.17 | 58.12 | 61.24 | 63.68 | 52.88 | 53.21 | 53.96 | 55.40 | 60.69 | 57.26 | 58.61 | 60.95 |
| | AS | 62.94 | 62.67 | 63.22 | 66.27 | 58.37 | 56.76 | 56.78 | 59.16 | 53.17 | 52.77 | 52.56 | 53.79 | 56.96 | 54.02 | 53.65 | 55.54 |
| | CLAWS | 71.11 | 70.76 | 77.32 | 78.66 | 66.07 | 64.48 | 66.29 | 69.34 | 61.15 | 60.82 | 61.13 | 63.52 | 63.75 | 60.69 | 64.97 | 67.89 |
| TabM | PPL | 44.17 | 44.84 | 44.73 | 47.61 | 49.54 | 47.44 | 57.60 | 47.80 | 44.17 | 44.14 | 43.89 | 43.50 | 48.89 | 45.42 | 60.63 | 46.06 |
| | WE | 59.94 | 59.40 | 51.84 | 61.08 | 54.28 | 53.64 | 61.87 | 55.35 | 51.44 | 51.03 | 49.40 | 52.96 | 54.54 | 52.81 | 63.59 | 53.08 |
| | LE | 52.78 | 52.51 | 48.32 | 55.00 | 52.50 | 51.56 | 59.79 | 51.76 | 47.48 | 47.23 | 43.22 | 45.68 | 51.77 | 50.04 | 65.28 | 52.06 |
| | HS | 65.03 | 64.90 | 63.72 | 71.21 | 60.35 | 58.89 | 69.00 | 63.68 | 54.66 | 54.59 | 50.40 | 55.40 | 60.25 | 57.57 | 70.61 | 60.95 |
| | AS | 62.90 | 62.26 | 57.64 | 66.27 | 58.20 | 57.27 | 64.66 | 59.16 | 50.65 | 49.97 | 48.80 | 53.79 | 55.84 | 53.76 | 64.66 | 54.94 |
| | CLAWS | 69.62 | 69.19 | 69.18 | 74.51 | 65.20 | 63.98 | 74.83 | 69.17 | 61.15 | 60.82 | 61.13 | 63.52 | 65.13 | 62.79 | 75.60 | 67.35 |

Table 14: Evaluation results for hallucination detection using OREAL-7B

| Dataset | | TEST | | | | AMC | | | | AIME | | | | A(J)HSME | | | |
|---|---|---|---|---|---|---|---|---|---|---|---|---|---|---|---|---|---|
| Strategy | Method | $F1_w$ | $F1_m$ | $AP_m$ | AUROC | $F1_w$ | $F1_m$ | $AP_m$ | AUROC | $F1_w$ | $F1_m$ | $AP_m$ | AUROC | $F1_w$ | $F1_m$ | $AP_m$ | AUROC |
| XGBOOST | PPL | 59.22 | 49.94 | 36.97 | 54.44 | 54.35 | 47.79 | 43.30 | 56.57 | 72.00 | 49.27 | 18.63 | 48.58 | 49.75 | 47.51 | 49.10 | 54.52 |
| | WE | 60.40 | 50.83 | 38.54 | 57.13 | 56.09 | 49.39 | 43.11 | 57.40 | **73.10** | 45.26 | 18.48 | 49.66 | 52.84 | 50.64 | 50.98 | 58.15 |
| | LE | 56.21 | 44.73 | 33.26 | 48.61 | 51.53 | 44.15 | 38.09 | 49.52 | 72.89 | 49.29 | 19.12 | 50.26 | 44.64 | 41.85 | 45.93 | 50.64 |
| | HS | **64.38** | 58.48 | 43.79 | 60.96 | 61.46 | **58.64** | **46.34** | **59.01** | 72.17 | **53.12** | **19.66** | 50.69 | **57.84** | **56.87** | **51.96** | 58.27 |
| | AS | 53.83 | 40.14 | 38.01 | 56.00 | 47.70 | 38.35 | 40.78 | 54.58 | 73.07 | 44.88 | 19.58 | **53.26** | 39.22 | 35.55 | 44.96 | 50.35 |
| | CLAWS | 64.31 | **58.82** | **44.59** | **62.34** | 57.95 | 55.58 | 44.36 | 57.67 | 70.94 | 50.29 | 19.47 | 48.47 | 53.84 | 53.17 | 47.38 | 53.54 |
| MLP | PPL | 63.58 | 57.86 | 58.36 | 60.87 | **61.39** | **57.22** | **60.84** | **63.86** | 72.77 | 50.52 | **54.10** | 57.11 | **57.81** | 56.53 | **62.58** | **63.95** |
| | WE | 53.83 | 40.14 | 53.34 | 55.76 | 47.70 | 38.35 | 50.38 | 49.42 | **73.07** | 44.88 | 49.57 | 48.51 | 39.22 | 35.55 | 46.93 | 43.33 |
| | LE | 53.83 | 40.14 | 48.33 | 46.47 | 47.70 | 38.35 | 47.30 | 45.62 | **73.07** | 44.88 | 48.70 | 45.18 | 39.22 | 35.55 | 50.10 | 50.16 |
| | HS | 59.85 | 56.95 | 59.13 | 61.23 | 57.76 | 56.68 | 59.33 | 60.93 | 65.99 | 49.85 | 51.27 | 51.93 | 55.95 | 55.86 | 58.06 | 59.23 |
| | AS | **66.27** | 61.13 | 60.03 | 63.33 | 59.15 | 56.22 | 55.25 | 57.60 | 66.15 | 50.58 | 51.16 | 52.63 | 57.78 | **57.30** | 58.53 | 60.12 |
| | CLAWS | 64.71 | **61.29** | **63.95** | **66.93** | 56.54 | 56.08 | 59.53 | 61.73 | 68.43 | 51.28 | 51.04 | 50.94 | 56.14 | 56.45 | 58.49 | 60.57 |
| TabM | PPL | 59.36 | 49.44 | 41.76 | 60.30 | 56.70 | 50.27 | 50.12 | **63.76** | 72.67 | 49.88 | **20.65** | 56.42 | 51.83 | 49.60 | **56.57** | **64.01** |
| | WE | 60.40 | 50.83 | 36.82 | 55.76 | 56.09 | 49.39 | 42.67 | 55.17 | **73.10** | 45.26 | 18.97 | 50.41 | 52.84 | 50.64 | 50.67 | 56.85 |
| | LE | 53.83 | 40.14 | 33.18 | 51.29 | 47.70 | 38.35 | 35.34 | 47.56 | 73.07 | 44.88 | 18.84 | 52.45 | 39.22 | 35.55 | 45.74 | 50.69 |
| | HS | 65.67 | 59.53 | **46.96** | 61.59 | 62.04 | **58.94** | **51.10** | 60.95 | 72.72 | **51.80** | 20.36 | 51.17 | **57.68** | 56.55 | 56.12 | 59.23 |
| | AS | 53.83 | 40.14 | 42.08 | 61.99 | 47.70 | 38.35 | 42.90 | 57.50 | 73.07 | 44.88 | 20.29 | 52.54 | 39.22 | 35.55 | 49.02 | 56.70 |
| | CLAWS | **66.82** | **61.04** | 46.92 | **63.98** | 60.38 | 57.71 | 44.02 | 58.87 | 72.66 | 51.27 | 19.93 | 51.23 | 57.67 | **56.93** | 52.52 | 59.28 |

Table 15: Evaluation results for hallucination detection using Qwen-2.5-Math-7B

| Dataset | | TEST | | | | AMC | | | | AIME | | | | A(J)HSME | | | |
|---|---|---|---|---|---|---|---|---|---|---|---|---|---|---|---|---|---|
| Strategy | Method | $F1_w$ | $F1_m$ | $AP_m$ | AUROC | $F1_w$ | $F1_m$ | $AP_m$ | AUROC | $F1_w$ | $F1_m$ | $AP_m$ | AUROC | $F1_w$ | $F1_m$ | $AP_m$ | AUROC |
| XGBOOST | PPL | 61.27 | 59.63 | 70.49 | 65.17 | 71.32 | 55.49 | 84.41 | 60.70 | 53.64 | 53.62 | **57.96** | **57.56** | 75.31 | 51.90 | 87.48 | 56.02 |
| | WE | 61.82 | 59.59 | 71.87 | 69.07 | **74.08** | 56.51 | 85.10 | 63.88 | 49.14 | 49.09 | 52.66 | 53.82 | **79.36** | **54.39** | 87.86 | 60.02 |
| | LE | 50.31 | 47.63 | 54.19 | 46.79 | 64.87 | 49.60 | 78.87 | 50.02 | 44.29 | 44.25 | 48.77 | 48.05 | 68.83 | 47.76 | 83.42 | 47.38 |
| | HS | 60.24 | 58.39 | 71.75 | 65.48 | 73.30 | 57.63 | 86.31 | 65.38 | 51.06 | 51.04 | 52.35 | 52.96 | 76.28 | 53.50 | 86.34 | 56.05 |
| | AS | 53.53 | 50.47 | 65.72 | 59.94 | 70.02 | 51.53 | 82.11 | 55.60 | 44.77 | 44.71 | 48.40 | 48.59 | 76.44 | 51.33 | 86.24 | 52.55 |
| | CLAWS | **65.68** | **64.26** | **76.73** | **71.49** | 70.33 | 55.37 | **87.57** | 64.43 | **54.20** | **54.19** | 56.73 | 56.92 | 74.70 | 53.65 | **89.57** | **61.67** |
| MLP | PPL | 66.79 | 66.76 | 69.52 | 73.18 | 72.67 | **62.12** | **65.22** | 70.57 | 59.03 | 59.04 | 65.01 | 65.82 | 76.35 | 58.21 | 58.34 | 65.10 |
| | WE | 48.63 | 44.28 | 56.64 | 58.59 | 68.80 | 50.49 | 53.17 | 57.00 | 49.56 | 49.56 | 50.83 | 50.02 | 63.09 | 47.25 | 52.70 | 53.42 |
| | LE | 41.52 | 38.90 | 45.78 | 40.81 | 54.04 | 42.87 | 47.37 | 43.32 | 39.42 | 39.40 | 43.82 | 40.81 | 60.20 | 42.24 | 47.40 | 41.58 |
| | HS | 67.34 | 66.26 | 68.94 | 72.35 | **73.89** | 59.54 | 61.08 | 68.01 | 52.22 | 52.24 | 53.83 | 54.20 | 74.35 | 54.64 | 54.44 | 58.05 |
| | AS | 59.31 | 58.50 | 61.44 | 62.87 | 61.52 | 50.92 | 53.04 | 55.88 | 48.03 | 48.02 | 49.07 | 47.88 | 64.78 | 47.88 | 51.44 | 53.56 |
| | CLAWS | **69.13** | **68.11** | **73.19** | **74.80** | 68.24 | 57.50 | 58.89 | 65.89 | 51.41 | 51.43 | 53.89 | 54.39 | **77.81** | 56.34 | 55.41 | 61.42 |
| TabM | PPL | 54.59 | 51.04 | 77.93 | 74.38 | 71.03 | 46.78 | **88.28** | 70.25 | 42.73 | 42.66 | **64.11** | **65.78** | 77.71 | 46.49 | **90.16** | 65.19 |
| | WE | 44.28 | 39.16 | 66.10 | 63.23 | 70.53 | 45.80 | 82.92 | 59.29 | 35.37 | 35.28 | 50.26 | 50.65 | 77.97 | 47.21 | 85.75 | 54.04 |
| | LE | 42.61 | 38.82 | 56.70 | 42.63 | 62.39 | 45.80 | 77.26 | 44.41 | 38.08 | 38.04 | 47.82 | 40.96 | 66.08 | 44.04 | 81.65 | 41.85 |
| | HS | **63.90** | **62.04** | 76.98 | 72.73 | 73.50 | 55.38 | 87.61 | 68.01 | 51.06 | 51.04 | 53.91 | 54.20 | 77.91 | 53.08 | 88.03 | 58.05 |
| | AS | 41.98 | 36.51 | 66.28 | 57.65 | 69.89 | 44.17 | 82.10 | 55.16 | 33.49 | 33.40 | 48.82 | 48.13 | 77.56 | 45.83 | 86.34 | 53.51 |
| | CLAWS | 63.38 | 61.74 | 77.83 | 71.17 | 69.70 | **55.45** | 87.38 | 63.99 | **53.38** | **53.38** | 56.58 | 56.14 | **78.89** | **53.44** | 89.54 | 61.99 |

Table 16: Evaluation results for creativity detection on a balanced dataset. The evaluation strategies used are Threshold and Prototype. Bold values indicate the best performance, underlined values indicate the second-best, and gray-shaded cells correspond to results where the model performed detection over only two out of the three target classes.

| Dataset | | TEST | | | | AMC | | | | AIME | | | | A(J)HSME | | | |
|---------|--------|--------|--------|--------|--------|--------|--------|--------|--------|--------|--------|--------|--------|--------|--------|--------|--------|
| Model | Method | $F1_w$ | $F1_m$ | $AP_m$ | AUROC | $F1_w$ | $F1_m$ | $AP_m$ | AUROC | $F1_w$ | $F1_m$ | $AP_m$ | AUROC | $F1_w$ | $F1_m$ | $AP_m$ | AUROC |
| Deepseek | PPL | 35.22 | 35.22 | 34.40 | 52.03 | **37.57** | **37.57** | 35.04 | 53.16 | **42.57** | **42.57** | **37.65** | **57.34** | 36.21 | 36.21 | 34.83 | 52.30 |
| | WE | 30.54 | 30.54 | 35.34 | 53.44 | 28.46 | 28.46 | 34.20 | 51.60 | 25.80 | 25.80 | 33.15 | 49.40 | 28.70 | 28.70 | 34.35 | 51.95 |
| | LE | 31.89 | 31.89 | 32.98 | 48.91 | 32.78 | 32.78 | 33.35 | 49.95 | 29.32 | 29.32 | 32.65 | 47.62 | 33.06 | 33.06 | 33.37 | 49.94 |
| | HS | 27.36 | 27.36 | 32.21 | 45.16 | 31.40 | 31.40 | 33.25 | 49.58 | 32.09 | 32.09 | 33.14 | 49.21 | 32.88 | 32.88 | 33.58 | 50.53 |
| | AS | 31.48 | 31.48 | 33.81 | 49.69 | 32.11 | 32.11 | 33.31 | 49.91 | 30.56 | 30.56 | 33.56 | 50.40 | 33.29 | 33.29 | 33.55 | 50.47 |
| | CLAWS | **46.30** | **46.30** | **41.34** | **62.03** | 35.90 | 35.90 | **35.87** | **54.62** | 36.93 | 36.93 | 36.66 | 55.95 | **36.43** | **36.43** | **35.97** | **54.89** |
| Mathstral | PPL | 29.11 | 29.11 | 32.86 | 48.70 | 32.37 | 32.37 | 33.39 | 49.88 | 27.67 | 27.67 | 32.46 | 47.79 | 31.55 | 31.55 | 33.04 | 49.29 |
| | WE | 38.76 | 38.76 | 36.14 | 54.71 | 36.23 | 36.23 | 35.05 | 53.11 | **34.21** | **34.21** | 33.81 | 50.74 | 34.19 | 34.19 | 34.41 | 52.05 |
| | LE | 30.60 | 30.60 | 32.83 | 48.54 | 32.05 | 32.05 | 32.99 | 49.08 | 26.00 | 26.00 | 31.56 | 44.85 | 30.59 | 30.59 | 32.62 | 48.09 |
| | HS | 27.80 | 27.80 | 33.82 | 49.35 | 26.85 | 26.85 | 33.12 | 48.96 | 19.96 | 19.96 | 31.70 | 45.22 | 25.67 | 25.67 | 32.73 | 48.38 |
| | AS | 24.86 | 24.86 | 31.88 | 44.16 | 28.19 | 28.19 | 32.16 | 46.26 | 28.62 | 28.62 | 32.91 | 48.53 | 29.09 | 29.09 | 32.16 | 46.82 |
| | CLAWS | **42.50** | **42.50** | **40.40** | **60.71** | **38.13** | **38.13** | **37.08** | **56.45** | 31.86 | 31.86 | **34.23** | **51.84** | **38.04** | **38.04** | **37.05** | **56.43** |
| OpenMath2 | PPL | 29.78 | 29.78 | 33.01 | 49.23 | 27.45 | 27.45 | 32.15 | 46.56 | 25.40 | 25.40 | 31.73 | 44.40 | 23.79 | 23.79 | 31.37 | 43.75 |
| | WE | 33.85 | 33.85 | 34.18 | 51.55 | 33.45 | 33.45 | 34.29 | 51.89 | 31.01 | 31.01 | 32.73 | 48.51 | 29.53 | 29.53 | 33.14 | 49.50 |
| | LE | 36.34 | 36.34 | 34.53 | 52.32 | **40.00** | **40.00** | 36.16 | 55.15 | 31.18 | 31.18 | 33.42 | 50.00 | **38.06** | **38.06** | 35.53 | 53.93 |
| | HS | 25.49 | 25.49 | 31.53 | 44.33 | 28.34 | 28.34 | 32.25 | 46.91 | 36.30 | 36.30 | 34.92 | 52.61 | 28.43 | 28.43 | 32.30 | 47.38 |
| | AS | 23.92 | 23.92 | 31.19 | 43.04 | 29.84 | 29.84 | 32.75 | 48.20 | **38.59** | **38.59** | **35.52** | **54.10** | 32.32 | 32.32 | 33.77 | 50.30 |
| | CLAWS | **41.90** | **41.90** | **38.92** | **58.51** | 37.66 | 37.66 | **36.93** | **56.36** | 24.86 | 24.86 | 33.22 | 49.63 | 33.47 | 33.47 | **35.60** | **54.23** |
| OREAL | PPL | 29.02 | 29.02 | 32.41 | 47.47 | 23.55 | 23.55 | 31.56 | 44.09 | 31.64 | 31.64 | 33.65 | 50.00 | 23.87 | 23.87 | 31.48 | 44.25 |
| | WE | 25.69 | 25.69 | 32.14 | 46.91 | 27.60 | 27.60 | 33.08 | 49.37 | 30.21 | 30.21 | 33.34 | 50.00 | 27.38 | 27.38 | 33.00 | 49.07 |
| | LE | **33.34** | **33.34** | **33.86** | **50.84** | **34.64** | **34.64** | 34.96 | 53.16 | 29.33 | 29.33 | 33.37 | 49.47 | 35.79 | 35.79 | 35.33 | 53.88 |
| | HS | 30.03 | 30.03 | 32.87 | 48.60 | 26.77 | 26.77 | 31.91 | 45.89 | 33.93 | 33.93 | 33.76 | 50.53 | 27.64 | 27.64 | 32.10 | 46.58 |
| | AS | 26.10 | 26.10 | 33.10 | 47.75 | 31.65 | 31.65 | 34.15 | 51.58 | 25.07 | 25.07 | 33.61 | 50.53 | 30.21 | 30.21 | 33.59 | 49.22 |
| | CLAWS | 25.27 | 25.27 | 32.99 | 48.31 | 34.08 | 34.08 | **35.16** | **53.48** | **34.85** | **34.85** | **34.69** | **52.66** | **37.49** | **37.49** | **35.52** | **54.04** |
| Qwen-2.5 | PPL | 27.04 | 27.04 | 34.14 | 50.00 | 27.75 | 27.75 | 33.72 | 49.52 | 25.76 | 25.76 | 33.34 | 49.53 | 35.59 | 31.09 | 33.92 | 50.47 |
| | WE | 34.62 | 34.62 | 34.91 | 52.96 | 32.83 | 32.83 | 34.20 | 51.39 | 31.12 | 31.12 | 33.01 | 49.06 | 29.08 | 28.56 | 33.01 | 49.11 |
| | LE | **45.25** | **45.25** | **39.53** | **59.24** | 40.54 | 40.54 | **36.60** | **55.60** | **39.56** | **39.56** | **36.05** | **54.56** | **41.31** | 39.10 | 35.66 | 54.32 |
| | HS | 27.84 | 27.84 | 34.67 | 51.72 | 30.39 | 30.39 | 35.66 | 53.58 | 18.97 | 18.97 | 33.48 | 50.31 | 36.80 | 31.55 | 35.08 | 53.11 |
| | AS | 26.88 | 26.88 | 32.66 | 47.17 | 31.59 | 31.59 | 34.13 | 51.27 | 32.32 | 32.32 | 34.20 | 51.73 | 37.59 | 34.97 | 34.28 | 51.52 |
| | CLAWS | 31.34 | 31.34 | 33.39 | 49.63 | **40.88** | **40.88** | **38.04** | **57.63** | 23.76 | 23.76 | 31.66 | 45.28 | 38.27 | 36.63 | 35.56 | 53.95 |

Table 17: Evaluation results for creativity detection using DeepSeek-Math-7B on a balanced dataset. Bold values indicate the best performance, underlined values indicate the second-best. Light gray-shaded cells correspond to cases where the model performed detection over only two out of the three target classes, while dark gray-shaded cells indicate cases where the model predicted only one out of the three target classes.

| Dataset | | TEST | | | | AMC | | | | AIME | | | | A(J)HSME | | | |
|---|---|---|---|---|---|---|---|---|---|---|---|---|---|---|---|---|---|
| Strategy | Method | F1$_w$ | F1$_m$ | AP$_m$ | AUROC | F1$_w$ | F1$_m$ | AP$_m$ | AUROC | F1$_w$ | F1$_m$ | AP$_m$ | AUROC | F1$_w$ | F1$_m$ | AP$_m$ | AUROC |
| XGBOOST | PPL | 39.12 | 39.12 | 37.18 | 55.02 | 35.85 | 35.85 | 35.79 | 53.32 | 36.68 | 36.68 | 35.70 | 52.40 | **35.90** | **35.90** | 35.24 | 52.73 |
| | WE | 33.20 | 33.20 | 40.18 | 60.71 | 30.24 | 30.24 | 37.15 | **56.72** | 22.32 | 22.32 | 34.51 | 51.56 | 31.12 | 31.12 | 36.72 | **56.17** |
| | LE | 36.50 | 36.50 | 35.10 | 52.32 | 34.29 | 34.29 | 34.37 | 51.49 | 32.17 | 32.17 | 33.29 | 49.80 | 33.12 | 33.12 | 34.87 | 52.15 |
| | HS | 36.52 | 36.52 | 37.35 | 54.52 | 35.71 | 35.71 | 34.28 | 51.47 | 35.93 | 35.93 | 37.00 | 54.95 | 35.13 | 35.13 | 35.19 | 51.92 |
| | AS | 41.05 | 41.05 | 41.24 | 59.10 | 34.01 | 34.01 | 34.79 | 51.88 | 31.78 | 31.78 | 37.18 | 53.95 | 32.39 | 32.39 | 33.68 | 50.52 |
| | CLAWS | 43.24 | 43.24 | 46.66 | 63.42 | 37.50 | 37.50 | 37.80 | 55.29 | 42.11 | 42.11 | 41.87 | 59.88 | 35.53 | 35.53 | 37.34 | 54.77 |
| MLP | PPL | 30.04 | 30.04 | 38.58 | 57.19 | 23.69 | 23.69 | 38.57 | 56.47 | 30.73 | 30.73 | 40.75 | 60.01 | 27.11 | 27.11 | 38.43 | 56.75 |
| | WE | 33.76 | 33.76 | 40.38 | 60.54 | 32.25 | 32.25 | 35.88 | 54.05 | 31.12 | 31.12 | 36.99 | 55.19 | 31.64 | 31.64 | 36.92 | 56.42 |
| | LE | 35.27 | 35.27 | 36.45 | 54.32 | 26.92 | 26.92 | 34.54 | 51.59 | 29.76 | 29.76 | 36.82 | 54.31 | 34.95 | 34.95 | 34.99 | 51.25 |
| | HS | 41.43 | 41.43 | 44.90 | 62.33 | 27.66 | 27.66 | 34.81 | 51.37 | 28.28 | 28.28 | 36.62 | 53.21 | 28.84 | 28.84 | 33.96 | 49.65 |
| | AS | 38.64 | 38.64 | 41.96 | 60.54 | 34.57 | 34.57 | 34.69 | 51.74 | 32.82 | 32.82 | 35.22 | 51.83 | 33.22 | 33.22 | 33.77 | 50.43 |
| | CLAWS | **44.98** | **44.98** | **49.40** | **67.35** | **41.38** | **41.38** | **43.04** | **60.77** | **41.75** | **41.75** | **42.96** | **62.88** | **35.00** | **35.00** | **42.14** | **60.25** |
| TabM | PPL | 34.50 | 34.50 | 38.91 | 57.59 | 33.63 | 33.63 | 38.62 | 56.63 | 33.89 | 33.89 | 41.36 | 59.97 | 36.75 | 36.75 | 38.43 | 56.73 |
| | WE | 30.20 | 30.20 | 40.38 | 60.66 | 29.00 | 29.00 | 37.21 | 56.72 | 25.12 | 25.12 | 36.46 | 54.90 | 29.53 | 29.53 | 36.91 | 56.41 |
| | LE | 37.40 | 37.40 | 35.83 | 54.04 | 33.79 | 33.79 | 34.81 | 51.50 | 39.61 | 39.61 | 37.83 | 55.14 | 35.52 | 35.52 | 35.20 | 52.10 |
| | HS | 40.49 | 40.49 | 44.16 | 60.99 | 32.64 | 32.64 | 35.74 | 52.89 | 34.37 | 34.37 | 35.28 | 52.06 | 30.71 | 30.71 | 33.73 | 49.32 |
| | AS | 41.17 | 41.17 | 42.68 | 60.83 | 34.17 | 34.17 | 34.72 | 51.63 | 32.61 | 32.61 | 37.28 | 54.15 | 32.51 | 32.51 | 33.70 | 50.42 |
| | CLAWS | **46.42** | **46.42** | 47.89 | **64.08** | 37.89 | 37.89 | 38.87 | 56.16 | **42.70** | **42.70** | **45.24** | **61.62** | **38.92** | **38.92** | **40.27** | **58.51** |

Table 18: Evaluation results for creativity detection using Mathstral-7B on a balanced dataset

| Dataset | | TEST | | | | AMC | | | | AIME | | | | A(J)HSME | | | |
|---|---|---|---|---|---|---|---|---|---|---|---|---|---|---|---|---|---|
| Strategy | Method | F1$_w$ | F1$_m$ | AP$_m$ | AUROC | F1$_w$ | F1$_m$ | AP$_m$ | AUROC | F1$_w$ | F1$_m$ | AP$_m$ | AUROC | F1$_w$ | F1$_m$ | AP$_m$ | AUROC |
| XGBOOST | PPL | 33.41 | 33.41 | 34.60 | 51.21 | 35.84 | 35.84 | 34.78 | 51.49 | 38.15 | 38.15 | 39.75 | **55.00** | 34.06 | 34.06 | 34.72 | 51.35 |
| | WE | 34.18 | 34.18 | 37.59 | 56.00 | 33.95 | 33.95 | 36.71 | 56.02 | 32.49 | 32.49 | 33.80 | 50.17 | 29.30 | 29.30 | 37.49 | 56.35 |
| | LE | 29.74 | 29.74 | 32.30 | 46.74 | 34.67 | 34.67 | 35.20 | 52.35 | 30.95 | 30.95 | 35.16 | 52.07 | 33.39 | 33.39 | 33.54 | 50.56 |
| | HS | 37.82 | 37.82 | 38.99 | 56.58 | 36.03 | 36.03 | 35.64 | 52.11 | 32.52 | 32.52 | 34.83 | 49.69 | 33.65 | 33.65 | 34.11 | 50.75 |
| | AS | 36.54 | 36.54 | 37.89 | 54.51 | 36.50 | 36.50 | 36.47 | 53.84 | 31.42 | 31.42 | 31.34 | 45.44 | 34.36 | 34.36 | 35.45 | 52.44 |
| | CLAWS | **40.20** | **40.20** | **45.46** | **60.50** | **40.14** | **40.14** | **40.23** | **57.46** | **42.73** | **42.73** | **41.03** | 54.54 | **34.60** | **34.60** | **38.38** | **56.42** |
| MLP | PPL | 29.20 | 29.20 | 41.65 | 59.58 | 16.67 | 16.67 | 35.93 | 52.16 | 16.48 | 16.48 | 36.69 | 53.09 | 27.26 | 27.26 | 35.73 | 52.53 |
| | WE | 27.15 | 27.15 | 36.88 | 51.56 | 16.67 | 16.67 | 35.44 | 52.37 | 22.70 | 22.70 | 31.40 | 45.42 | 19.64 | 19.64 | 33.94 | 49.79 |
| | LE | 37.59 | 37.59 | 37.39 | 54.32 | 30.70 | 30.70 | 35.27 | 53.14 | 32.48 | 32.48 | 37.23 | 52.87 | 16.67 | 16.67 | 33.33 | 50.00 |
| | HS | 45.58 | 45.58 | 43.92 | 62.70 | 29.61 | 29.61 | 37.66 | 55.65 | 19.92 | 19.92 | 36.00 | 52.67 | 29.42 | 29.42 | 35.91 | 53.20 |
| | AS | 38.49 | 38.49 | 42.73 | 62.29 | 36.60 | 36.60 | 36.38 | 54.85 | 31.81 | 31.81 | 35.14 | 52.07 | 25.42 | 25.42 | 35.28 | 53.28 |
| | CLAWS | **46.78** | **46.78** | 45.64 | 63.92 | **44.79** | **44.79** | **44.69** | **62.51** | 33.72 | 33.72 | 43.75 | 58.92 | **39.02** | **39.02** | **41.95** | **60.36** |
| TabM | PPL | 42.24 | 42.24 | 38.69 | 55.69 | 36.79 | 36.79 | 35.46 | 51.75 | 37.77 | 37.77 | 36.86 | 53.20 | 34.14 | 34.14 | 35.34 | 52.11 |
| | WE | 31.48 | 31.48 | 36.95 | 51.88 | 30.15 | 30.15 | 35.69 | 52.18 | 31.55 | 31.55 | 32.46 | 47.29 | 22.68 | 22.68 | 33.75 | 49.64 |
| | LE | 28.89 | 28.89 | 36.88 | 53.46 | 29.67 | 29.67 | 35.14 | 52.70 | 26.54 | 26.54 | 37.27 | 52.38 | 28.46 | 28.46 | 34.87 | 52.11 |
| | HS | 34.64 | 34.64 | 39.16 | 58.56 | 31.46 | 31.46 | 37.67 | 55.25 | 20.70 | 20.70 | 33.21 | 47.77 | 28.15 | 28.15 | 35.51 | 52.67 |
| | AS | 38.21 | 38.21 | 40.78 | 58.34 | 36.79 | 36.79 | 37.88 | 55.80 | 31.80 | 31.80 | 34.94 | 50.89 | 35.48 | 35.48 | 35.89 | 53.82 |
| | CLAWS | **45.74** | **45.74** | **48.76** | **66.40** | **41.22** | **41.22** | **43.57** | **61.17** | **38.58** | **38.58** | **44.41** | **60.86** | **38.99** | **38.99** | **40.32** | **57.89** |

Table 19: Evaluation results for creativity detection using OpenMath2-LLaMA3.1-8B on a balanced dataset

| Dataset | | TEST | | | | AMC | | | | AIME | | | | A(J)HSME | | | |
|---|---|---|---|---|---|---|---|---|---|---|---|---|---|---|---|---|---|
| Strategy | Method | $F1_w$ | $F1_m$ | $AP_m$ | AUROC | $F1_w$ | $F1_m$ | $AP_m$ | AUROC | $F1_w$ | $F1_m$ | $AP_m$ | AUROC | $F1_w$ | $F1_m$ | $AP_m$ | AUROC |
| XGBOOST | PPL | 31.64 | 31.64 | 34.48 | 50.89 | 29.55 | 29.55 | 32.14 | 48.11 | 35.35 | 35.35 | 38.34 | 53.75 | 35.58 | 35.58 | 37.00 | 53.21 |
| | WE | 37.97 | 37.97 | 39.76 | 57.89 | 34.03 | 34.03 | 35.96 | 54.12 | 34.70 | 34.70 | 35.60 | 51.49 | 29.54 | 29.54 | 35.44 | 53.48 |
| | LE | 31.75 | 31.75 | 34.09 | 49.66 | 33.41 | 33.41 | 34.72 | 51.37 | 32.31 | 32.31 | 35.01 | 51.03 | 33.04 | 33.04 | 34.44 | 50.80 |
| | HS | 32.30 | 32.30 | 35.72 | 53.62 | 37.06 | 37.06 | 38.54 | 55.35 | 32.31 | 32.31 | 35.01 | 51.03 | 33.38 | 33.38 | 35.13 | 52.06 |
| | AS | 36.01 | 36.01 | 37.54 | 54.58 | 35.51 | 35.51 | 34.71 | 51.48 | 31.35 | 31.35 | 31.95 | 47.58 | 33.35 | 33.35 | 33.55 | 51.22 |
| | CLAWS | 49.96 | 49.96 | 48.76 | 66.23 | 42.00 | 42.00 | 42.66 | 60.32 | 36.91 | 36.91 | 37.91 | 54.72 | 40.50 | 40.50 | 42.87 | 60.46 |
| MLP | PPL | 29.07 | 29.07 | 35.24 | 50.63 | 29.70 | 29.70 | 33.56 | 49.09 | 23.99 | 23.99 | 34.08 | 48.25 | 31.77 | 31.77 | 42.10 | 59.40 |
| | WE | 32.54 | 32.54 | 34.84 | 50.68 | 28.22 | 28.22 | 34.47 | 50.89 | 16.67 | 16.67 | 34.71 | 51.53 | 31.38 | 31.38 | 36.24 | 53.50 |
| | LE | 31.28 | 31.28 | 34.74 | 49.56 | 25.55 | 25.55 | 32.21 | 47.23 | 18.22 | 18.22 | 30.51 | 43.57 | 28.08 | 28.08 | 35.97 | 52.38 |
| | HS | 33.89 | 33.89 | 43.40 | 61.26 | 29.16 | 29.16 | 39.11 | 56.93 | 34.24 | 34.24 | 37.61 | 51.56 | 28.82 | 28.82 | 36.98 | 54.43 |
| | AS | 40.56 | 40.56 | 41.38 | 60.35 | 27.80 | 27.80 | 36.07 | 52.92 | 27.55 | 27.55 | 31.38 | 46.84 | 23.34 | 23.34 | 30.92 | 45.34 |
| | CLAWS | 49.51 | 49.51 | 49.09 | 68.00 | 43.34 | 43.34 | 46.49 | 64.34 | 27.91 | 27.91 | 43.29 | 60.89 | 43.42 | 43.42 | 43.28 | 60.58 |
| TabM | PPL | 23.49 | 23.49 | 33.31 | 47.71 | 24.11 | 24.11 | 32.64 | 47.23 | 21.31 | 21.31 | 33.87 | 46.70 | 23.14 | 23.14 | 34.57 | 49.12 |
| | WE | 36.25 | 36.25 | 37.61 | 55.32 | 34.02 | 34.02 | 35.06 | 52.59 | 31.37 | 31.37 | 35.68 | 53.01 | 36.77 | 36.77 | 35.67 | 52.81 |
| | LE | 32.59 | 32.59 | 34.74 | 50.21 | 32.19 | 32.19 | 35.17 | 50.86 | 21.50 | 21.50 | 28.69 | 40.00 | 32.75 | 32.75 | 32.78 | 49.01 |
| | HS | 38.79 | 38.79 | 42.21 | 60.36 | 30.87 | 30.87 | 37.02 | 54.82 | 30.62 | 30.62 | 36.79 | 50.53 | 30.88 | 30.88 | 36.98 | 53.80 |
| | AS | 40.56 | 40.56 | 41.36 | 59.98 | 33.06 | 33.06 | 35.25 | 51.71 | 29.54 | 29.54 | 31.96 | 47.37 | 32.42 | 32.42 | 33.62 | 49.69 |
| | CLAWS | 45.87 | 45.87 | 50.47 | 69.09 | 41.45 | 41.45 | 47.00 | 63.95 | 41.43 | 41.43 | 44.73 | 59.39 | 40.84 | 40.84 | 44.47 | 61.44 |

Table 20: Evaluation results for creativity detection using OREAL-7B on a balanced dataset

| Dataset | | TEST | | | | AMC | | | | AIME | | | | A(J)HSME | | | |
|---|---|---|---|---|---|---|---|---|---|---|---|---|---|---|---|---|---|
| Strategy | Method | $F1_w$ | $F1_m$ | $AP_m$ | AUROC | $F1_w$ | $F1_m$ | $AP_m$ | AUROC | $F1_w$ | $F1_m$ | $AP_m$ | AUROC | $F1_w$ | $F1_m$ | $AP_m$ | AUROC |
| XGBOOST | PPL | 33.63 | 33.63 | 34.65 | 51.86 | 38.34 | 38.34 | 37.58 | 54.99 | 36.88 | 36.88 | 34.53 | 50.26 | 35.93 | 35.93 | 37.37 | 53.25 |
| | WE | 24.25 | 24.25 | 33.50 | 49.75 | 25.34 | 25.34 | 36.10 | 53.91 | 28.33 | 28.33 | 35.06 | 52.47 | 25.40 | 25.40 | 36.88 | 54.24 |
| | LE | 33.58 | 33.58 | 34.26 | 50.43 | 32.49 | 32.49 | 33.80 | 50.07 | 36.88 | 36.88 | 36.89 | 49.59 | 32.76 | 32.76 | 34.25 | 51.18 |
| | HS | 32.32 | 32.32 | 33.35 | 48.58 | 35.14 | 35.14 | 36.56 | 54.71 | 28.70 | 28.70 | 34.12 | 50.62 | 35.73 | 35.73 | 35.48 | 52.58 |
| | AS | 27.82 | 27.82 | 33.13 | 48.70 | 33.44 | 33.44 | 34.73 | 51.62 | 36.34 | 36.34 | 34.41 | 50.94 | 29.16 | 29.16 | 34.10 | 50.26 |
| | CLAWS | 39.05 | 39.05 | 40.73 | 59.03 | 34.99 | 34.99 | 36.47 | 53.69 | 34.14 | 34.14 | 36.99 | 50.62 | 35.84 | 35.84 | 36.48 | 53.33 |
| MLP | PPL | 30.55 | 30.55 | 39.80 | 56.69 | 37.16 | 37.16 | 41.26 | 59.01 | 22.47 | 22.47 | 41.00 | 54.61 | 40.38 | 40.38 | 40.95 | 56.56 |
| | WE | 16.71 | 16.71 | 32.98 | 49.07 | 18.17 | 18.17 | 34.96 | 52.38 | 26.00 | 26.00 | 35.41 | 52.23 | 20.66 | 20.66 | 34.70 | 50.55 |
| | LE | 25.62 | 25.62 | 37.46 | 55.19 | 16.67 | 16.67 | 36.41 | 53.56 | 27.58 | 27.58 | 37.50 | 54.35 | 16.67 | 16.67 | 33.09 | 48.46 |
| | HS | 26.31 | 26.31 | 34.00 | 50.79 | 33.04 | 33.04 | 39.84 | 57.51 | 24.03 | 24.03 | 41.83 | 55.60 | 35.34 | 35.34 | 38.18 | 56.24 |
| | AS | 16.67 | 16.67 | 34.39 | 49.49 | 29.69 | 29.69 | 35.08 | 51.92 | 23.90 | 23.90 | 32.64 | 46.50 | 34.91 | 34.91 | 41.78 | 60.46 |
| | CLAWS | 37.54 | 37.54 | 38.59 | 54.61 | 27.03 | 27.03 | 38.90 | 55.98 | 30.64 | 30.64 | 33.68 | 47.74 | 39.49 | 39.49 | 40.95 | 56.25 |
| TabM | PPL | 36.95 | 36.95 | 42.66 | 59.17 | 38.03 | 38.03 | 40.84 | 57.92 | 37.60 | 37.60 | 40.37 | 54.71 | 39.07 | 39.07 | 42.25 | 57.69 |
| | WE | 21.91 | 21.91 | 32.71 | 47.72 | 22.22 | 22.22 | 33.44 | 50.59 | 25.16 | 25.16 | 33.97 | 50.85 | 20.66 | 20.66 | 34.26 | 51.71 |
| | LE | 32.76 | 32.76 | 34.28 | 50.15 | 33.43 | 33.43 | 34.97 | 52.29 | 34.93 | 34.93 | 38.28 | 52.85 | 36.14 | 36.14 | 37.31 | 55.85 |
| | HS | 27.86 | 27.86 | 34.51 | 50.49 | 36.48 | 36.48 | 39.10 | 56.42 | 31.06 | 31.06 | 36.33 | 52.99 | 34.38 | 34.38 | 38.41 | 56.11 |
| | AS | 31.94 | 31.94 | 35.29 | 51.14 | 34.14 | 34.14 | 35.55 | 52.82 | 33.61 | 33.61 | 34.50 | 50.43 | 29.68 | 29.68 | 37.19 | 55.81 |
| | CLAWS | 44.21 | 44.21 | 44.09 | 62.58 | 34.23 | 34.23 | 37.20 | 53.25 | 29.62 | 29.62 | 33.35 | 46.67 | 35.21 | 35.21 | 36.83 | 53.68 |

Table 21: Evaluation results for creativity detection using Qwen-2.5-Math-7B on a balanced dataset

| Dataset | | TEST | | | | AMC | | | | AIME | | | | A(J)HSME | | | |
|---|---|---|---|---|---|---|---|---|---|---|---|---|---|---|---|---|---|
| Strategy | Method | $F1_w$ | $F1_m$ | $AP_m$ | AUROC | $F1_w$ | $F1_m$ | $AP_m$ | AUROC | $F1_w$ | $F1_m$ | $AP_m$ | AUROC | $F1_w$ | $F1_m$ | $AP_m$ | AUROC |
| XGBOOST | PPL | 40.16 | 40.16 | **43.09** | **60.01** | 37.57 | 37.57 | 38.43 | 55.33 | **37.99** | **37.99** | **39.15** | **56.60** | 36.22 | 35.16 | 36.07 | 52.68 |
| | WE | 41.38 | **41.38** | 40.81 | 58.87 | 37.89 | 37.89 | 38.51 | 56.66 | 35.50 | 35.50 | 36.17 | 52.77 | 37.33 | 36.20 | 36.72 | 54.31 |
| | LE | 33.93 | 33.93 | 33.71 | 51.48 | 32.65 | 32.65 | 34.00 | 50.92 | 29.51 | 29.51 | 32.34 | 48.28 | 34.55 | 33.51 | 34.05 | 50.89 |
| | HS | 39.46 | 39.46 | 40.97 | 59.22 | 38.29 | 38.29 | 39.60 | 57.37 | 33.19 | 33.19 | 33.33 | 49.91 | 38.94 | **37.34** | 36.09 | 53.49 |
| | AS | 36.63 | 36.63 | 37.97 | 54.72 | 38.21 | 38.21 | 35.99 | 53.15 | 30.86 | 30.86 | 33.03 | 47.91 | 38.05 | 35.96 | 34.76 | 52.47 |
| | CLAWS | **42.08** | 36.63 | 37.97 | 54.72 | **39.02** | **39.02** | **39.92** | **58.99** | 35.24 | 35.24 | 37.61 | 54.24 | **39.06** | 37.01 | **38.00** | **57.28** |
| MLP | PPL | 43.21 | 43.21 | **48.48** | **67.07** | 38.35 | 38.35 | **45.18** | **63.38** | **36.12** | **36.12** | **41.57** | **60.18** | 37.57 | **38.13** | 40.67 | 58.45 |
| | WE | 23.73 | 23.73 | 35.91 | 52.21 | 35.18 | 35.18 | 35.83 | 52.69 | 32.29 | 32.29 | 35.18 | 51.72 | 32.97 | 32.69 | 34.81 | 50.79 |
| | LE | 20.88 | 20.88 | 29.52 | 40.91 | 27.44 | 27.44 | 34.87 | 50.93 | 22.47 | 22.47 | 30.83 | 44.62 | 27.76 | 26.99 | 32.09 | 46.89 |
| | HS | 42.56 | 42.56 | 45.23 | 62.86 | 31.55 | 31.55 | 40.19 | 59.28 | 32.19 | 32.19 | 35.77 | 52.09 | 33.65 | 32.88 | 38.15 | 54.69 |
| | AS | 39.11 | 39.11 | 37.77 | 56.80 | 26.66 | 26.66 | 36.45 | 54.02 | 26.01 | 26.01 | 31.89 | 47.33 | 34.46 | 33.71 | 34.39 | 51.12 |
| | CLAWS | **43.56** | **43.56** | 46.59 | 64.77 | **38.36** | **38.36** | 41.98 | 61.53 | 33.57 | 33.57 | 37.93 | 55.03 | **38.59** | 35.12 | **41.92** | **60.47** |
| TabM | PPL | 40.13 | 40.13 | **47.62** | **66.36** | 39.52 | 39.52 | **43.87** | **62.15** | 34.74 | 34.74 | **41.71** | **59.99** | 35.79 | 36.55 | **39.37** | 57.42 |
| | WE | 38.23 | 38.23 | 38.37 | 55.63 | 35.77 | 35.77 | 37.37 | 54.90 | 30.03 | 30.03 | 35.10 | 51.88 | 32.93 | 32.90 | 35.90 | 52.83 |
| | LE | 26.16 | 26.16 | 30.58 | 46.48 | 29.14 | 29.14 | 32.54 | 50.05 | 27.55 | 27.55 | 31.08 | 46.65 | 28.17 | 27.31 | 32.96 | 48.78 |
| | HS | 40.42 | 40.42 | 44.41 | 61.94 | 38.08 | 38.08 | 41.85 | 59.30 | 34.03 | 34.03 | 35.72 | 52.50 | 36.22 | 34.52 | 36.96 | 53.35 |
| | AS | 36.75 | 36.75 | 38.01 | 56.30 | 36.24 | 36.24 | 36.48 | 53.82 | 29.81 | 29.81 | 33.21 | 48.37 | 34.77 | 33.98 | 34.55 | 51.32 |
| | CLAWS | **41.14** | **41.14** | 44.03 | 61.98 | **40.16** | **40.16** | 40.10 | 59.94 | **36.52** | **36.52** | 37.83 | 54.64 | **40.03** | **37.98** | 38.50 | **57.69** |

