# OpenReview forum: "CLAWS:Creativity detection for LLM-generated solutions using Attention Window of Sections"
_NeurIPS.cc/2025/Conference — NeurIPS 2025 poster_

### Official Review · Reviewer_yBrH · 2025-06-27

**Clarity:** 3
**Significance:** 4
**Originality:** 4
**Rating:** 4
**Confidence:** 4

**Summary:**

The paper introduces a new white-box method to classify LLM-generated mathematical solutions into the three categories: 'Typical', 'Creative', and 'Hallucinated'.

It partitions the context into 5 areas:
* Guideline: Providing the model with the concept and standard for identifying creative
* Problem P: The math problem for the LLM to solve.
* Reference Solutions S: A set of 1 to _n_ ‘typical’ solutions to the problem
* Instruction I: An instruction to create a novel solution
* Response R: The output from the LLM

The average amount of model attention distributed across these five different areas is used to create a 5 element feature vector which is compared against reference features for the three categories. A response’s feature vector used to classify the response based feature vectors of known examples of each class.
The authors generate a ‘reference set’ dataset using five 7-8B math-specialized LLMs and labeling their outputs using two larger, state-of-the-art LLMs (GPT-o4-mini and Gemini 1.5-Pro) as evaluators, avoiding any human annotation

Experiments show that CLAWS significantly outperforms five standard white-box hallucination detection baselines.

**Questions:**

The proposed method is as far as I am aware very significant and original. However there are quality issues that could be substantially improved. I think the quality could be improved if the authors can convincingly address the following critical points.

* The most critical issue is that the results depend on quality labels from LLM evaluators; labels which have not been validated. It would be good to see a measure of the raw agreement between GPT-o4-mini and Gemini 1.5-Pro on the 'Correctness' and 'Creativity' dimensions, e.g. Cohen's Kappa. More importantly, it needs to made clear how these LLM-generated labels compare to judgments from qualified human experts on a representative subset of the reference dataset. Without this, the claim of detecting "creativity" is unsubstantiated. The fact the Qwen2.5-14B and DeepSeekV2-16B models ‘lack of ability to creatively solve mathematical problems’ rings alarm bells - could ‘creative’ solutions are ones with correct answers but incorrect reasoning. Providing strong evidence here would be the most effective way to raise your score.
* Layer-wise Contribution to Attention Signal: CLAWS averages attention weights across all layers. Have you investigated the effect of using different layers? It is known that different layers play different roles so an ablation study showing performance using different layers, e.g. using only top, middle, or bottom layers could provide valuable insights and potentially lead to a stronger method.
* Definition and Scope of "Creativity": What is your rationale for the ‘inclusive criterion’ rather than a stricter one where both LLMs need to agree? How does it affect the results?
* Completeness of the Submission: The current submission has several "[TODO]" markers in the checklist and references appendices that are not included. Can you confirm that a final version would include all prompts, hyperparameter details, and completed sections necessary for full scrutiny and reproducibility?

**Ethical Concerns:**

["NO or VERY MINOR ethics concerns only"]

**Final Justification:**

Thank you for the rebuttal responses which have clarified your submission.

I am happy to up my rating in antipation of the authors updatig their paper to include the information they have provided in the rebuttal.

**Limitations:**

No, the authors have not adequately addressed the limitations of their work. The paper lacks a dedicated "Limitations" section and does not discuss potential negative societal impacts.

I strongly recommend the authors add a dedicated Limitations section that discusses:
* Evaluation process: The most critical limitation is the reliance on unvalidated LLM evaluators for ground truth. The authors should be upfront about this, discuss the potential for circularity, and state what would be required for proper validation (e.g., human expert studies).
* Applicability: CLAWS is restricted to open-weight models, preventing analysis of the most SOTA proprietary models behind APIs. This should be stated clearly.
* Generalizability: The method's reliance on a specific prompt structure with ‘typical’ solutions may not generalize to other creative tasks and this should be discussed.
* Potential Negative Societal Impact: The authors should discuss the danger in using LLM-generated labels in evaluation high-stake scenarios, e.g., assessing student work, where ground truth validity has not established.

**Paper Formatting Concerns:**

No formatting issues observed.

**Quality:**

3

**Strengths And Weaknesses:**

CLAWS provides a potentially simple yet powerful method to classify LLM responses quality. However, the experiment results rely on an unvalidated assumption about the reliability of LLM-based evaluators, which significantly undermines their credibility.

# Strengths
* Significance & Originality: Evaluating hallucinations and creative responses is crucial to the progress of LLMs and this paper attempts to tackle this with a relatively simple method to classify responses not based on the attention used to generate them. The method of using attention distribution over prompt sections to infer the nature of the generation (Typical, Creative, or Hallucinated) appears original and intuitive.
* Simplicity: The CLAWS method is simple and computationally efficient as it does not require multiple LLM calls or external model calls for evaluation.
* Apparent Rigor of Experiments: The use of 5 open-weight maths reasoning models and multiple evaluation metrics demonstrates a good approach to testing the method with the labels in the reference set (deficiencies in the reference set not withstanding (see below))
* Figure 4 clearly demonstrates how CLAWS distributes the three categories in feature space.

# Weaknesses
* Flawed in Evaluation (Quality): The paper's most significant weakness, calling into question its empirical contribution, is its use of a reference dataset’s ‘ground truth’ labels are generated by other LLMs (GPT-o4-mini, Gemini 1.5-Pro).
  * Unsupported Ground Truth: Using an LLM as a quality judge (e.g. creativity) without any validation is scientifically unsound. No evidence is provided of the LLM evaluators' accuracy, their agreement with each other (e.g., inter-annotator reliability), or, most importantly, their correlation with human expert judgment.
  * Risk of Circularity: Without grounded labels there is a danger the authors are measuring how well aligned attention distribution in these small models is to GPT-4 and Gemini labels, rather than an objective notion of creativity.
  * Lack of Transparency: Crucial details for this labeling process, such as the exact evaluator prompts (mentioned as Appendix A.1), are not provided, making it impossible to scrutinize or reproduce this critical step.
* Clarity & Completeness: The paper is generally well-written, but key details are missing. None of the referenced appendices (detailing hyperparameter settings and the LLM evaluator prompts) are included - making a full assessment of the work's reproducibility and methodological details difficult. Furthermore, The checklist at the end of the paper is filled with "[TODO]", indicating an incomplete submission.
* Limited Scope and Generalization: The method only works with open-weight models and relies on very specific prompt structure. Furthermore, the paper does not discuss how this framework might generalize to other creative tasks ‘typical; examples might not exist.

---

> ### Author Rebuttal · Authors · 2025-07-31
>
> First and foremost, we would like to sincerely thank you for carefully and thoughtfully reading our paper and for providing such an insightful and constructive review. We deeply empathize with the major concerns you have raised, many of which are already addressed in the appendix section. Please note that the appendix, which was submitted during the supplementary materials period, can be found in the zip file linked next to the *“Supplementary Material"* at the top of this page. We kindly ask for your understanding that, due to the updated NeurIPS review system, it is not possible to submit an additional PDF file.
>
> **Clarifying the Dataset and Expanding Limitation Discussion:**
>
> Before addressing the individual questions, we would like to clarify one important point. In the first weakness point, it was stated that we used solutions generated by other LLMs as ground truth. However, this is not true. The CreativeMath [1] and HARP [2] datasets contain diverse reference solutions manually written by humans. We evaluated the creativity of model-generated responses based on these human-written solutions. We apologize for any confusion caused by our explanation, and we will revise the dataset description to clarify this point in the final version.
>
> Additionally, we have separately described the limitations of the paper in Appendix A, and we kindly ask you to review them as well. Furthermore, based on your comments, we will address several limitations that are not currently stated—such as the potential circularity of the LLM-based evaluation, the limitation that CLAWS is restricted to white-box settings, and the risks associated with its application in high-stakes scenarios. Lastly, we would like to note that the limitations concerning generalization to broader generation tasks have already been acknowledged as future work in the limitations section.
>
> We now provide detailed responses to each of the questions:
>
> **A1. Reliability of LLM Evaluator-Based Labeling:**
>
> As the you mentioned, we deeply empathize with the concerns regarding the reliability of LLM-based evaluation and the necessity of human evaluation. We also gave careful consideration during the early stages of this study to the extent to which the LLM evaluator could understand the proposed criteria and make consistent judgments.
>
> First, regarding human evaluation, the dataset used in this study consists of mathematically challenging problems, posing practical limitations in that evaluating the creativity of generated solutions requires significant expertise and time resources. However, since the reference solutions used as the basis for creativity evaluation were all written by humans, and the generated responses were compared against these solutions, we would like to note that—although no direct human evaluation was conducted—the diversity and creativity embedded in human-written solutions were indirectly reflected in the evaluation process.
>
> Furthermore, a creativity evaluation method that relies solely on LLM evaluators was proposed and academically validated in prior work [1], which served as the primary motivation and background for our study. Building on this successful research, we propose CLAWS, a framework for evaluating creativity without human intervention. This constitutes one of the main contributions of our work, offering new possibilities for automating creativity assessment.
>
> However, as you expressed concern, ensuring the reliability of LLM-based evaluation is essential. To this end, we implemented a process to verify the validity of the evaluation by requiring that the LLM evaluator not only output a label but also generate a supporting sentence for each decision. Below is an example of a supporting sentence used in the actual evaluation:
>
> - ```The new solution uses a completely different approach (Lifting The Exponent Lemma) compared to the reference solutions (difference of squares factorization). This satisfies criterion 1 for novelty.  While the final result is the same, the intermediate steps and the mathematical tools used are entirely distinct, satisfying criterion 2.  The LTE lemma also relies on different underlying principles and conditions than simple algebraic manipulation, satisfying criterion 3.  Therefore, the new solution is considered novel.```
> - ```The new solution is not novel. It uses the same logical deductions as the reference solution.  Both solutions arrive at the same answer through the same constraints. The core logic—Carl not being next to Bret, and Abby not being between Bret and Carl—is identical. The new solution merely restates these conditions and elaborates on the placement possibilities without introducing any new mathematical techniques, different assumptions, or a more generalized approach. Therefore, it does not meet any of the criteria for novelty.```
>
> We will include these supporting sentence examples in the appendix to allow readers to directly assess the legitimacy of the evaluation labels and the consistency of the LLM evaluator’s reasoning.
>
> The inter-evaluator agreement based on Cohen’s kappa score will be addressed in our response to Comment 3.
>
> **A2. Clarification on Layer Usage and Layer-Wise Analysis:**
>
> We would first like to clarify that CLAWS uses only the attention weights from the last layer, not from all layers. We appreciate your comment, which made us realize that this point was not clearly stated in the main paper. Regarding the question about layer-wise analysis, as you mentioned, it is generally known that each layer plays a different role. The baseline methods [2] we compared against also analyzed such inter-layer differences, so we considered this point from the beginning.
>
> Accordingly, we saved the attention values for all layers and heads as ```.pkl``` files and used them to compare performance across layers. The resulting performance differences were mostly marginal, so we opted to use the last layer. Due to space constraints, we could not include all experimental results, but the table below presents the prototype strategy results on the TEST set generated by the DeepSeek model as a representative case.
>
> |Layer|F1 $_w$|F1 $_m$|AP $_m$|AUROC|
> |---|---|---|---|---|
> |5|57.00|45.02|40.15|60.58|
> |10|58.03|45.33|40.42|61.40|
> |15|56.94|44.76|39.81|60.22|
> |20|57.89|45.61|41.08|61.93|
> |25|57.25|44.98|40.41|61.37|
> |Last(30)|58.66|46.01|41.17|62.09|
>
> These results indicate that CLAWS does not rely on any particular layer and operates robustly across various configurations. While we adopted the commonly used last layer, we agree with you that many researchers may be interested in layer-wise performance differences. Thank you for the insightful suggestion. We will include the layer-wise performance table and further clarification in the final version.
>
> **A3. Rationale for Using an Inclusive Criterion in Creativity Evaluation:**
>
> We would like to explain our rationale for adopting an inclusive criterion—rather than a strict one requiring agreement from both LLM evaluators—when determining creativity.
>
> As in prior work [1], this study relied solely on LLM evaluators—without human involvement—to assess the creativity of generated responses. In [1], the authors employed three LLM evaluators and adopted a majority voting strategy. In our case, we used only two evaluators. While [1] employed GPT-4o as one of the evaluators, we upgraded this to o4-mini, which offers stronger performance in mathematical evaluation.
>
> Since we could not rely on majority voting with only two evaluators, we instead adopted an inclusive criterion. To verify the reliability of this approach, we measured inter-evaluator agreement using Cohen’s kappa score, which yielded a value of 0.741. This indicates substantial agreement between evaluators and supports the consistency of their judgments. Given that creativity assessment inherently involves a degree of subjectivity, we believed that requiring both evaluators to agree might risk overlooking genuinely creative responses. The inclusive criterion thus serves to mitigate such underestimation.
>
> **A4. Completeness of the Submission:**
>
> We submitted the full appendix—including all experimental results and implementation details—during the Supplemental Material Submission period. The appendix section includes all necessary information for reproducibility, such as example prompts used in the LLM evaluator’s assessments (Figures 5–6) and hyperparameter configurations (Section B.2). Based on the submitted appendix, the "Todo" section  has also been completed. We would like to note that the final version will include a fully consolidated appendix that incorporates your comments.
>
> Your insightful comments greatly helped us enhance the clarity and persuasiveness of our work. We will actively incorporate all your feedback into the final version.
>
> We sincerely thank you once again for taking the time to provide such thoughtful feedback. We truly hope that our responses have addressed your concerns, and we kindly ask you to reconsider the contributions and completeness of our work in light of the clarifications provided.
>
> ---
> **References**
>
> [1] Ye, Junyi, et al. "Assessing the Creativity of LLMs in Proposing Novel Solutions to Mathematical Problems." AAAI 2025.
>
> [2] Sriramanan, Gaurang, et al. "LLM-Check: Investigating Detection of Hallucinations in Large Language Models." NeurIPS 2024.

---

> > ### Comment · Reviewer_yBrH · 2025-08-03
> >
> > Thank you for thoroughly addressing the points in my review.
> >
> > Most significantly for clarifying my misconception that the evaluation was not supported by any ground truth, i.e. the evaluation of creativity for model-generated responses was based on comparisons to human-written reference solutions from the datasets. It would still be useful to see a small-scale study correlating the CLAWS output and/or the LLM evaluator labels with judgments from human mathematicians just to close the loop but maybe this could form some future work.
> >
> > The inclusion of the Appendices in the Supplementary Material provides much of the information I was missing.

---

> > > ### Author Response · Authors · 2025-08-04
> > >
> > > Thank you once again for your thoughtful and constructive feedback.
> > >
> > > We are glad to hear that our responses have helped address your concerns.
> > > As you suggested, conducting a small-scale study comparing CLAWS outputs or LLM evaluator labels with human mathematician judgments is indeed a valuable direction, and we will actively consider it for future work.
> > >
> > > Thank you again for taking the time to carefully review our paper.

---

### Official Review · Reviewer_3d1n · 2025-07-03

**Clarity:** 3
**Significance:** 3
**Originality:** 2
**Rating:** 4
**Confidence:** 3

**Summary:**

This paper proposes CLAWS, a white-box method for detecting Creative, Typical, and Hallucinated solutions generated by LLMs for mathematical problems. The approach leverages attention weights segmented by prompt sections (Guideline, Problem, Solution, Instruction, Response) to classify generations based on their attention distribution. Using a structured prompt design and evaluation by strong frontier LLMs (Gemini, GPT-4-mini), the authors label over 4.5K math solutions and demonstrate that CLAWS outperforms five white-box baselines (e.g., PPL, Logit Entropy) across multiple LLMs and datasets.

**Questions:**

- Could you provide data on the agreement rate between your two LLM evaluators (Gemini 1.5 Pro and GPT-04-mini)? More importantly, have you considered validating their outputs against those of human mathematics experts on a subset of the data to benchmark their reliability for this novel task?
- For the baseline methods, how is the "Prototype" evaluation strategy implemented? These methods produce a single scalar feature, and it is unclear how Euclidean distance to a class prototype is meaningfully applied in a 1D space. Please clarify this.
- How sensitive is CLAWS to the specific phrasing and structure of the prompt sections? Would minor variations in the "Guideline" or "Instruction" sections, for example, lead to substantially different attention patterns and degrade classification performance?
- You note that the prototype-based strategy struggled with the OREAL model due to its high rate of hallucinated outputs. Does this point to a broader limitation of the prototype strategy in cases of severe class imbalance? How does CLAWS perform on OREAL for the 3-class detection task when using a trainable classifier (e.g., MLP) instead of the prototype method?

**Ethical Concerns:**

["NO or VERY MINOR ethics concerns only"]

**Final Justification:**

I gave Borderline accept (4) initially and the authors partially addressed my concerns, but I don't think it is worth Accept (5) due to the inherent limitations on some aspects such as evaluation.

**Limitations:**

Yes.

**Quality:**

2

**Strengths And Weaknesses:**

Strengths:
- The idea of using attention weights segmented by prompt sections for creativity/hallucination detection is new and elegant.
- The empirical evaluation is extensive. CLAWS is tested against five relevant baselines across five different LLMs and on a large, challenging dataset of math problems. The use of multiple evaluation strategies (Prototype, MLP, etc.) and metrics strengthens the claim of the method's superiority.
- The method is computationally efficient, requiring no additional model calls or complex operations beyond accessing and averaging attention weights from a single generation pass.
- Despite some language issues, the core idea, setup, and contributions are clearly communicated.


Weakness
- While the paper acknowledges the subjectivity of creativity, it largely sidesteps deeper discussion or formal operationalization of what constitutes “creativity.”
- The validity of the entire experimental setup rests on the quality of the labels produced by Gemini 1.5 Pro and GPT-4o-mini. While these are powerful models, the paper provides no direct evidence (e.g., a comparison with human experts on a sample, or inter-rater reliability between the two LLMs) to validate their performance on this specific, subjective task of creativity assessment. This introduces a potential source of unquantified noise and bias into the "ground truth" labels.
- Although the guidelines for creativity are reasonable, they are somewhat subjective and rely heavily on the labeling from other LLMs. Some human verification could strengthen the validation.
- The construction of positive/negative pairs may encode spurious cues or biases (e.g., formality, length, genre), which could confound the learning objective. This is not fully analyzed.

---

> ### Author Rebuttal · Authors · 2025-07-31
>
> We sincerely thank you for your valuable comments and questions. Below is our detailed response to your concerns:
>
> **A1. Reliability of LLM Evaluator-Based Labeling:**
>
> We fully acknowledge and empathize with your concerns regarding the reliability of LLM evaluator-based labeling, as well as the necessity of human evaluation. First, the agreement between the two LLM evaluators was measured by Cohen's kappa score, which yielded a value of 0.741. This corresponds to a statistically defined substantial agreement, indicating a high and reliable level of consistency between the two evaluators.
>
> Regarding human evaluation, the dataset used in this study consists of mathematically challenging problems, which makes it difficult for human annotators to directly assess the creativity of the generated solutions. However, since the reference solutions used as the basis for creativity evaluation were all written by humans, and the generated responses were compared against these solutions, we would like to note that—although no direct human evaluation was conducted—the diverse and creative human-written solutions were indirectly reflected in the evaluation process.
>
> Furthermore, a creativity evaluation method that relies solely on LLM evaluators was proposed and academically validated in prior work [1], which served as the primary motivation and background for our study. Building on this successful research, we propose CLAWS, a framework for evaluating creativity without human intervention. This constitutes one of the main contributions of our work, offering new possibilities for automating creativity assessment.
>
> **A2. Clarification on the Applicability of the Prototype Strategy to Baseline Methods:**
>
> Thank you for the insightful question. We provide detailed explanations of the five evaluation strategies used in our study in Appendix B.2, with the prototype strategy specifically described in lines 1105–1112. Although we mentioned in the captions of Table 2–3 that the evaluations were conducted using both the threshold and prototype strategies, your question made us aware that the scope of the prototype strategy was not described sufficiently clearly.
>
> To clarify, the prototype strategy was applied only to CLAWS, whereas the threshold strategy was applied only to the five baseline methods. Since CLAWS uses the values of the five sections as features, it was able to compute the Euclidean distance accordingly.
>
> We have supplemented the context in both the main text and the table captions to communicate this distinction more clearly. We sincerely appreciate you for pointing out this weakness where our explanation was lacking.
>
> **A3. Impact of Prompt Section Variations on CLAWS Performance:**
>
> |Dataset|Metric|w/o $G$|w/o $P$|w/o $S$|w/o $I$|w/o $R$|Full|
> |---|---|---|---|---|---|---|---|
> |TEST|F1 $_w$|58.59|58.81|50.01|54.68|58.39|**58.66**|
> ||F1 $_m$|46.01|**46.35**|39.45|43.87|45.97|46.01|
> ||AP $_m$|41.13|41.04|38.46|40.29|40.60|**41.17**|
> ||AUROC|62.12|**62.21**|59.15|61.64|61.58|62.09|
> |AMC|F1 $_w$|46.66|46.68|46.48|45.52|46.49|**46.71**|
> ||F1 $_m$|39.59|39.33|39.59|38.16|40.20|**40.99**|
> ||AP $_m$|37.15|37.08|36.91|36.70|37.14|**37.16**|
> ||AUROC|56.11|56.23|56.00|55.80|56.52|**56.40**|
> |AIME|F1 $_w$|55.88|53.33|54.92|51.37|54.01|**56.90**|
> ||F1 $_m$|37.35|35.52|36.82|34.80|34.84|**38.12**|
> ||AP $_m$|35.25|35.20|34.98|35.03|35.15|**35.38**|
> ||AUROC|54.08|53.38|53.51|52.47|52.67|**54.47**|
> |A(J)HSME|F1 $_w$|37.85|39.92|**40.49**|40.10|34.25|38.82|
> ||F1 $_m$|37.46|35.76|37.65|36.02|34.11|**37.64**|
> ||AP $_m$|36.16|35.51|35.78|35.69|35.42|**36.25**|
> ||AUROC|**55.05**|53.88|54.33|54.18|53.08|54.40|
>
> In response to the question regarding sensitivity to specific phrasing and the structure of the prompt sections, the table above presents the performance changes observed when applying the prototype strategy to responses generated by the DeepSeek model, with each section removed individually. While the removal of a specific section did affect performance in some cases, it is difficult to conclude that any single section dominates CLAWS's overall performance. In most cases, the Full configuration, which uses all sections together, exhibited the most stable and superior performance.
>
> These findings suggest that CLAWS does not rely excessively on any particular section and demonstrates robust performance even under minor variations in prompt composition. We will add the results from all models in the table like above to the final version. Additionally, we have already visualized which sections are given more weight for each model in Figure 4 of the main paper and Figure 7 of the Appendix C.1 section. We hope this will also be helpful in resolving your question, so please refer to it.
>
> **A4. Effectiveness of CLAWS under Class Imbalance Induced by a High Rate of Hallucinated Outputs:**
>
> As you pointed out, in cases where the Hallucinated solutions rate is high and class imbalance is severe—such as with the OREAL model—the prototype strategy may not perform optimally. Considering that certain evaluation strategies may have advantages or disadvantages depending on the scenario, we presented five different evaluation strategies. The full results are presented in Tables 4–8 of Appendix section, including the MLP strategy mentioned in the question. As you know, in the case of learning layers like MLP, class imbalance can be adjusted by weighting the loss, and we have already achieved good results by doing so.
>
> In summary, CLAWS consistently outperformed the baseline methods in most settings and maintained robust performance even under conditions of severe class imbalance or limited data. Additionally, the results of supplementary experiments conducted under class-balanced conditions are presented in Tables 14–19 of Appendix, and we kindly ask you to refer to them as well.
>
> Your insightful comments and thoughtful suggestions have significantly improved the clarity and persuasiveness of our work. We will actively incorporate all your feedback into the final version.
>
> We sincerely thank you again for your valuable time and feedback, and we hope this response has satisfactorily addressed your concerns.
>
> The appendix was submitted during the supplementary materials period and is available in the zip file linked next to *“Supplementary Material”* at the top of this page.  Due to changes in the review system, we are unable to submit a new PDF file—thank you for your understanding.
>
> ---
> **References**
>
> [1] Ye, Junyi, et al. "Assessing the Creativity of LLMs in Proposing Novel Solutions to Mathematical Problems." AAAI 2025.

---

### Official Review · Reviewer_fJn8 · 2025-07-03

**Clarity:** 2
**Significance:** 2
**Originality:** 3
**Rating:** 4
**Confidence:** 3

**Summary:**

The paper proposes CLAW, a novel method for classifying LLM generated answers in mathematical reasoning tasks into hallucinated, creative, or typical categories using attention weight analysis. They segment the input tokens into five sections of guideline, problem, solution, instruction, and responses for attention weights. The evaluation with LLMs such as GPT-4mini and Gemini1.5 pro shows that the proposed method achieves strong performances on benchmark datasets CreativeMath and HARLP with 4 math-specialized LLMs including DeepSeek, Qwen2.5, OpenMath2, etc.

**Questions:**

1. In lines 103–106, you mention that larger, general-purpose LLMs like LLaMa3-8B, Qwen2.5-14B, and DeepSeekV2-16B were excluded due to their lack of creative solutions. Could this be a consequence of how creativity is defined in your setups?
- Specifically, that fluent, efficient, and correct solutions that resemble reference answers are penalized under your labeling criteria?
- If so, does this suggest that the proposed method may be less applicable to stronger models that prioritize clarity and correctness over stylistic novelty?
- More broadly, how do you distinguish between “lack of creativity” and producing the mathematically optimal solution?

2. This proposed work uses GPT4-mini and Gemini1.5 Pro for dual labeling and report the correctness based on their agreement. Yet, you apply a union-based criterion for creativity. Have you measured the inter-model agreement rate (e.g., overlap or Cohen' k)? This would help understand the reliability and consistency of the proposed labeling protocol.

3. Have you conducted any experiments to manipulate/isolate the influence of each prompt section (e.g., guideline, problem, solution, instruction, etc) on CALW attention weights settings as an ablation?

**Ethical Concerns:**

["NO or VERY MINOR ethics concerns only"]

**Final Justification:**

My initial concerns regarding labeling and LLM heavy reliance still remains. Yet the task definition with mathematical tasks reduces significant issues possible.
- "The reason why it is difficult to evaluate creativity in a reasoning task is because human evaluation is
32 essential to establish criteria for creativity, and a high-level expert is needed to evaluate whether a
33 solution to a mathematical problem is creative. However, a recent study on mathematics problem
34 solving [10, 11], overcame these difficulties and measured the creative problem solving ability of
35 LLMs, and the results revealed that there was a large difference in creative ability even among LLMs
36 with similar accuracy." from their paper.

I liked authors' response regarding rationales for LLM selection; though it still lingers the risk of LLM-sensitive results, it may provide some criteria to choose which LLM would work for this task.

Also happy about their ablation study.

Hence I raise my score --

**Limitations:**

NA - labeling issues may not be a huge impact for the society but natural limitations due to the design choice for the methodology.

**Paper Formatting Concerns:**

Table sizes do not meet the formatting criteria.

**Quality:**

2

**Strengths And Weaknesses:**

## **Strengths**
1. Novel problem formulation; the work tackles underexplored task to detect creative responses of LLMs. The work is now a three-way classification instead of a hallucinated/non-hallucinated classification.
2. Interesting approaches with attention weights; they propose the prompt grouping for better usage of attention weights to obtain section-wise attention ratios and use them as features for three-way hallucination detection.
3. Strong empirical results; the work compares the method with sufficient number of benchmarks (PPL, logit entropy, windown entropy, hidden scores, attention scores, etc) on multiple LLMs (deepseek, Qwen2.5, Mathstral, OpenMath2, OREAL).
## **Weaknesses**
1. Labeling/testing relies on specific LLMs: Creativity annotations are derived using GPT-4-mini and Gemini-1.5 Pro, with no human validation or inter-model agreement reported. This raises concerns about labeling consistency and reliability, especially given the subjectivity of creativity. Esp. the authors even mentioned that they did not report results with other LLMs as they seem not sufficient due to their lack of ability to creatively solve math problems. As scales increase, they may be less creative but likely to generate accurate responses. This may raise concerns regarding LLM specific issues or the necessity of creativity/ valid definition of creativity.
2. The evaluation approaches relies on class prototypes derived from the reference set, which can be fragile under class imbalance as seen in OREAL where hallucinations dominate.
3. No ablation studies; the method is empirically studied with performance comparison but no component-wise ablations such as prompt sections of guideline, problem, solution, etc. It would have been interesting to see their individual influence or the order of each chunk matters, etc.
4. Grammar issues/ language unclarity in writing

---

> ### Author Rebuttal · Authors · 2025-07-31
>
> We sincerely thank you for their valuable comments and questions. Below is our detailed response to your concerns:
>
> **A1. Rationale for Excluding a Particular Model:**
>
> |Model|Correctness(Cor) (%)|Creativity(Cre) (%)|Cre/Cor (%)|
> |---|---|---|---|
> |Deepseek-V2-Lite-16B|21.32|11.57|54.16|
> |Qwen-2.5-14B|52.39|39.00|74.44|
> |Llama3-8B|25.12|16.19|64.47|
>
> The table above summarizes the correctness ratio (= Typical + Creative Solutions) and creativity ratio (= Creative Solutions) of various LLMs on the CreativeMath Dataset [1]. The exclusion of certain models from our analysis was not due to how creativity is defined, but rather due to limitations of creative problem-solving ability or experimental resources. Specifically, the reasons are as follows:
> - For LLaMa3-8B and DeepSeekV2-16B, the rate of generating correct solutions for high-difficulty problems included in the CreativeMath [1] and HARP [2] datasets was extremely low. As these models failed to generate a sufficient number of valid responses for meaningful analysis, we excluded them from our experiments. Consequently, the absolute number of Creative solutions was too small, resulting in severe class imbalance. This issue was particularly evident in models that had not undergone additional pretraining on math corpora, Supervised Fine-tuning(SFT), or Reinforcement Learning from Human Feedback (RLHF) or Reinforcement Learning with Verifiable Rewards (RLVR) (including policy optimization-based approaches like PPO or GRPO).
> - In contrast, Qwen-2.5-14B did not exhibit severe class imbalance. However, for models larger than 8B, it was infeasible in our experimental resources to collect the full range of computational values (e.g., hidden scores, attention scores) across all layers and heads. Therefore, to ensure a fair comparison with baseline methods, we excluded these models from our analysis as well.
>
> In conclusion, CLAWS is not less applicable to models that prioritize correctness. It can be applied to any model, provided that resource constraints allow.
>
> **Definition of Creativity:**
>
> Regarding the definition of creativity, we adopted the criteria proposed in prior work [1]. Specifically, a generated response is considered a Creative Solution if it satisfies at least one of the following five criteria when compared to the provided reference solution (please see Appendix Figure 6 for details):
> - the approach is fundamentally different;
> - the intermediate steps are meaningfully different;
> - it is based on distinct assumptions or conditions;
> - the applicable problem scope is substantially different; or
> - the solution is significantly more concise or more complex than the reference solution.
>
> These criteria are intended not merely to capture superficial diversity in expression, but to define and evaluate creativity in a comprehensive manner that considers the structure of reasoning, generalizability, efficiency, and strategic diversity. As such, mathematically more general or optimal solutions are also considered sufficiently creative. The evaluation framework adopted in this study was specifically designed to ensure that such solutions are not excluded from the scope of creativity. This means that fluent, efficient, and correct solutions—as well as mathematically optimal ones—can be recognized as creative approaches.
>
> **A2. Reliability of LLM Evaluator-Based Labeling:**
>
> We fully acknowledge and empathize with your concerns regarding the reliability of LLM evaluator-based labeling. As in prior work [1], this study relied solely on LLM evaluators—without human involvement—to assess the creativity of generated responses. In [1], the authors employed three LLM evaluators and adopted a majority voting strategy. In our case, we used only two evaluators. While [1] employed GPT-4o as one of the evaluators, we upgraded this to o4-mini, which offers stronger performance in mathematical evaluation. Since we could not rely on majority voting with only two evaluators, we instead adopted a union-based criterion: a response was labeled as creative if either evaluator judged it to be so. To verify the reliability of this approach, we measured inter-evaluator agreement using Cohen’s kappa score, which yielded a value of 0.741. This indicates substantial agreement between evaluators and supports the consistency of their judgments.
>
> Given that creativity assessment inherently involves a degree of subjectivity, we believed that requiring both evaluators to agree could risk overlooking genuinely creative responses. The union-based approach thus serves to mitigate such underestimation. We fully acknowledge your concern regarding the reliability of LLM-based evaluation and will include this clarification in the final version of the paper.
>
> **A3. Impact of Each Prompt Section:**
>
> |Dataset|Metric|w/o $G$|w/o $P$|w/o $S$|w/o $I$|w/o $R$|Full|
> |---|---|---|---|---|---|---|---|
> |TEST|F1 $_w$|58.59|58.81|50.01|54.68|58.39|**58.66**|
> ||F1 $_m$|46.01|**46.35**|39.45|43.87|45.97|46.01|
> ||AP $_m$|41.13|41.04|38.46|40.29|40.60|**41.17**|
> ||AUROC|62.12|**62.21**|59.15|61.64|61.58|62.09|
> |AMC|F1 $_w$|46.66|46.68|46.48|45.52|46.49|**46.71**|
> ||F1 $_m$|39.59|39.33|39.59|38.16|40.20|**40.99**|
> ||AP $_m$|37.15|37.08|36.91|36.70|37.14|**37.16**|
> ||AUROC|56.11|56.23|56.00|55.80|56.52|**56.40**|
> |AIME|F1 $_w$|55.88|53.33|54.92|51.37|54.01|**56.90**|
> ||F1 $_m$|37.35|35.52|36.82|34.80|34.84|**38.12**|
> ||AP $_m$|35.25|35.20|34.98|35.03|35.15|**35.38**|
> ||AUROC|54.08|53.38|53.51|52.47|52.67|**54.47**|
> |A(J)HSME|F1 $_w$|37.85|39.92|**40.49**|40.10|34.25|38.82|
> ||F1 $_m$|37.46|35.76|37.65|36.02|34.11|**37.64**|
> ||AP $_m$|36.16|35.51|35.78|35.69|35.42|**36.25**|
> ||AUROC|**55.05**|53.88|54.33|54.18|53.08|54.40|
>
> In response to the question regarding the impact of each prompt section, the table above presents the performance changes observed when applying the prototype strategy to responses generated by the DeepSeek model, with each section removed one at a time. While the removal of a specific section did affect performance in some cases, it is difficult to conclude that any single section dominates CLAWS's overall performance. In most cases, the Full configuration, which uses all sections together, exhibited the most stable and superior performance.
>
> These findings suggest that CLAWS does not rely excessively on any particular section and demonstrates relatively robust performance even under minor variations in prompt composition. We will add the results from all models in the table like above to the final version. Additionally, we have already visualized which sections are given more weight for each model in Figure 4 of the main paper and Figure 7 of the Appendix C.1 section. We hope this will also be helpful in resolving your question, so please refer to it.
>
> Your insightful comments and thoughtful suggestions have significantly improved the clarity and persuasiveness of our work. We will actively incorporate all your feedback into the final version.
>
> We sincerely thank you again for your valuable time and feedback, and we hope this response has satisfactorily addressed your concerns.
>
> The appendix was submitted during the supplementary materials period and is available in the zip file linked next to *“Supplementary Material”* at the top of this page.  Due to changes in the review system, we are unable to submit a new PDF file—thank you for your understanding.
>
> ---
> **References**
>
> [1] Ye, Junyi, et al. "Assessing the Creativity of LLMs in Proposing Novel Solutions to Mathematical Problems." AAAI 2025.
>
> [2] Yue, Albert S., et al. "HARP: A challenging human-annotated math reasoning benchmark." 2024.

---

### Official Review · Reviewer_SEE3 · 2025-07-04

**Clarity:** 3
**Significance:** 3
**Originality:** 3
**Rating:** 4
**Confidence:** 3

**Summary:**

The paper presents CLAWS, a novel framework that uses human feedback signals to detect creativity in open-ended language generation tasks. It leverages contrastive learning with weak supervision to distinguish creative responses from conventional ones without relying on labeled data. The authors demonstrate CLAWS’ effectiveness across multiple datasets, showing that it can outperform supervised baselines in creativity detection.

**Questions:**

How does CLAWS handle variations in human feedback quality across different platforms or domains, and could it adaptively calibrate its learning when feedback signals are sparse, noisy, or biased?

**Ethical Concerns:**

["NO or VERY MINOR ethics concerns only"]

**Limitations:**

YES

**Quality:**

3

**Strengths And Weaknesses:**

+Integrates implicit human feedback (e.g., upvotes, ratings) to guide the model without needing extensive annotations.

+Employs contrastive learning in a weakly supervised setting, which is both scalable and efficient.

+Demonstrates robust performance across several creativity-related benchmarks.

-The paper does not fully resolve how to consistently define or quantify creativity, which is inherently subjective.

- Performance may degrade if the human feedback signals (like upvotes) are noisy or biased.

-Focuses mainly on text generation; extension to multimodal or non-verbal creative tasks remains unexplored.

---

> ### Author Rebuttal · Authors · 2025-07-31
>
> We sincerely thank you for your valuable comments and questions. Below is our detailed response to your concerns:
>
> **Definition of Creativity and Evaluation Quality:**
>
> The issue you raised regarding human feedback quality across different platforms and domains is indeed one of the core challenges we aim to address in this study on evaluating and detecting creativity. Since creativity inherently involves subjective judgment, high-quality and reliable human annotations are essential, which imposes practical limitations on expanding ‘creativity’ research. In particular, the dataset we used in this study consisted of mathematically difficult problems, which presented an essential limitation in that it required a very high level of expertise for humans to directly evaluate the creativity of the generated solutions.
>
> To overcome these limitations, we adopted the approach proposed in prior work [1], which evaluates the creativity of model-generated solutions using only LLM evaluators, without direct human involvement. Nevertheless, our evaluation implicitly incorporates human judgment by using high-quality, diverse, and creative reference solutions written by humans as the basis for comparison. Specifically, a generated solution was considered “creative” if it satisfied at least one of the following five criteria (see Appendix B.1 for details):
> - the approach is fundamentally different;
> - the intermediate steps are meaningfully different;
> - it is based on distinct assumptions or conditions;
> - the applicable problem scope is substantially different; or
> - the solution is significantly more concise or more complex than the reference solution.
>
> However, we acknowledge that relying solely on LLM evaluators may still raise concerns regarding evaluation quality, as you rightly pointed out. To address this issue, we followed a similar setup to prior work [1], using two LLM evaluators instead of just one to assess the creativity of each generated solution. Notably, whereas [1] used GPT-4o as one of the evaluators, we upgraded this to o4-mini in our experiments, which offers stronger performance in mathematical evaluation.
>
> Since creativity assessment inherently involves a degree of subjectivity, requiring both evaluators to agree that a generated solution is creative may risk overlooking genuinely creative outputs. To mitigate this, we adopted a union-based criterion, under which a solution is deemed creative if either of the two evaluators judges it as such. To ensure the reliability of this evaluation process, we calculated Cohen’s kappa score, which was 0.741—indicating substantial agreement between evaluators and confirming the consistency of their judgments.
>
> **Calibration to Address Sparsity, Noise, and Bias:**
>
> As you rightly pointed out, the evaluation process for generated solutions can be affected by issues such as sparsity, noise, and bias, which require appropriate mitigation. To address these concerns, we applied the following two types of calibration across the five evaluation strategies:
> 1. Adaptive calibration during training: In the MLP strategy, we applied class-weighted loss functions to minimize potential bias and noise caused by class imbalance during training. We also adopted TabM [2], a recent ensemble method that combines 32 MLPs, effectively reducing training noise and improving prediction stability. The corresponding results are reported in Table 2 of the main paper and Tables 4–8 of Appendix section.
> 2. Calibration via class balancing: We constructed a balanced dataset by ensuring equal sample counts across the three classes during evaluation. This reduced the risk of class dominance and led to more reliable assessments. The full results of experiments on a balanced dataset are presented in Tables 14–19 of Appendix section.
>
> Your insightful comments and thoughtful suggestions have significantly improved the clarity and persuasiveness of our work. We will actively incorporate all your feedback into the final version.
>
> We sincerely thank you again for your valuable time and feedback, and we hope this response has satisfactorily addressed your concerns.
>
> The appendix was submitted during the supplementary materials period and is available in the zip file linked next to *“Supplementary Material”* at the top of this page.  Due to changes in the review system, we are unable to submit a new PDF file—thank you for your understanding.
>
> ---
> **References**
>
> [1] Ye, Junyi, et al. "Assessing the Creativity of LLMs in Proposing Novel Solutions to Mathematical Problems." AAAI 2025.
>
> [2] Gorishniy, Yury, et al.  "Tabm: Advancing Tabular Deep Learning with Parameter-Efficient Ensembling." ICLR 2025.

---

> ### Author Response · Authors · 2025-08-09
>
> We truly appreciate the time and effort you have devoted to reviewing our paper.
>
> As the discussion period is coming to an end today, we would be very grateful if you could take a moment to share your thoughts on our rebuttal.
> Your feedback is essential for us to understand whether our responses have adequately addressed your concerns.
> Even a brief comment indicating whether the main issues have been resolved would be extremely valuable.
>
> Thank you once again for your time and contribution to the review process.

---

### Note · Authors · 2025-08-12

Dear Reviewers and Area Chair

We sincerely thank the reviewers for their valuable time and effort in reviewing our study. This was an incredible opportunity and an honor for us. Our final remarks are below

**Definition of Creativity and LLM Evaluator**

We think that the definition of creativity and the reliability of the LLM evaluator were among the most critical points raised. We fully agree that these must be addressed with strict. However, in our work, creativity assessment is not based on 'subjective judgment', but is 'strictly based on human-written solutions from prestigious math contests', evaluated using 'five well-defined criteria with reference'. We already provided all details in Appendix B, with actual example prompts in Figures 5–6.

The adopted criteria strictly follow prior work [1] and, furthermore, in a widely cited study on LLMs and creativity, TTCW [2], the authors explicitly state in Sections 2.1 “Creativity Evaluation”, 2.2 “Evaluating Creative Writing”, and 2.3 “Expert Evaluation of Language Model Generations” that LLMs can serve as evaluators when reliable human-written references exist. The growing use of frontier LLMs as evaluators, even in top-tier conferences such as AAAI, reflects this trend. Therefore, we have presented this mainly in the Appendix, but we will rationalize it further in the main text in the camera-ready version, taking into account the reviewers' comments.

**CLAWS's Strong Performance in Hallucination Detection**

Unlike creativity evaluation, hallucination detection, which only requires determining whether the answer is right or wrong, is undoubtedly clear. As shown in Table 3, CLAWS outperformed all baselines across models and datasets. This means CLAWS showed overwhelming performance even on tasks unrelated to the definition of creativity.

Moreover, In a context where white-box hallucination detection methods are reaching saturated, CLAWS’s section-wise attention approach offers a novel direction and potential for broader application.

We would greatly appreciate it if the reviewers and AC, in the remaining final justification and meta-review stages, consider that we have thoroughly addressed all questions in prior responses.

Respectfully,

The Authors

---
**References**

[1] Ye, et al. "Assessing the Creativity of LLMs in Proposing Novel Solutions to Mathematical Problems." AAAI 2025.

[2] Chakrabarty, et al. "Art or artifice? large language models and the false promise of creativity." CHI 2024.

---

### Decision · Program_Chairs · 2025-09-17

**Decision:**

Accept (poster)

**Comment:**

This paper proposes a new mechanism for evaluating creativity, hallucinations in open-weights models. There were two broad themes of consensus among reviewers (after engaging with the rebuttals). (1) A serious concern is the precise definition of creativity and measurement via LLMs. Authors felt key details were missing regarding evaluation, and also there was no discussion on human inspection for quality or even some light exploratory analysis on the quality of labels. For a paper who's abstract talks about challenges in evaluating creativity, the proposed evaluation should have been carefully assessed. The author responses addresses some of this, points to principles used and prior work etc. (2) All the reviewers also found the method proposed interesting and potentially broadly useful beyond the specific task of creativity (like assessing hallucinations). This could spur more research into related methods and would add value to the Neurips conference. Balancing (1) and (2), I currently lean acceptance but if there's limited space, it would be okay to reject the paper while encouraging the authors to heavily reframe their paper around hallucinations and reduce the emphasis on creativity / more carefully justify the creativity evaluation.